# V-InFoR: A Robust Graph Neural Networks Explainer for Structurally Corrupted Graphs

**Senzhang Wang**[*][†]
Central South University
szwang@csu.edu.cn

**Jun Yin**[†]
Central South University
yinjun2000@csu.edu.cn

**Chaozhuo Li**
Microsoft Research Asia
cli@microsoft.com

**Xing Xie**
Microsoft Research Asia
xing.xie@microsoft.com

**Jianxin Wang**
Central South University
jxwang@mail.csu.edu.cn

## Abstract

GNN explanation method aims to identify an explanatory subgraph which contains the most informative components of the full graph. However, a major limitation of existing GNN explainers is that they are not robust to the structurally corrupted graphs, e.g., graphs with noisy or adversarial edges. On the one hand, existing GNN explainers mostly explore explanations based on either the raw graph features or the learned latent representations, both of which can be easily corrupted. On the other hand, the corruptions in graphs are irregular in terms of the structural properties, e.g., the size or connectivity of graphs, which makes the rigorous constraints used by previous GNN explainers unfeasible. To address these issues, we propose a robust GNN explainer called V-InfoR [3]. Specifically, a robust graph representation extractor, which takes insights of variational inference, is proposed to infer the latent distribution of graph representations. Instead of directly using the corrupted raw features or representations of each single graph, we sample the graph representations from the inferred distribution for the downstream explanation generator, which can effectively eliminate the minor corruption. We next formulate the explanation exploration as a graph information bottleneck (GIB) optimization problem. As a more general method that does not need any rigorous structural constraints, our GIB-based method can adaptively capture both the regularity and irregularity of the severely corrupted graphs for explanation. Extensive evaluations on both synthetic and real-world datasets indicate that V-InfoR significantly improves the GNN explanation performance for the structurally corrupted graphs.

## 1 Introduction

Nowadays, graph data is ubiquitous in various domains, such as citation networks [1, 2, 3], social networks [4, 5], and chemical molecules [6]. GNN has manifested promising performance in many graph tasks, e.g., node classification [7, 8], link prediction [9], and graph classification [10, 11], by aggregating node features in light of the topological structures. Similar to other deep learning models, GNN models also have the defect that they are non-transparent, and the prediction results lack human-intelligible explanations [12]. Precise explanations can not only make people better understand GNN predictions but also assist GNN designers in detecting underlying flaws and purifying them.

---

[*]Corresponding author.

[†]Equally contribute to this research.

[3]Code and dataset are available at https://anonymous.4open.science/r/V-InfoR-EF88

37th Conference on Neural Information Processing Systems (NeurIPS 2023).

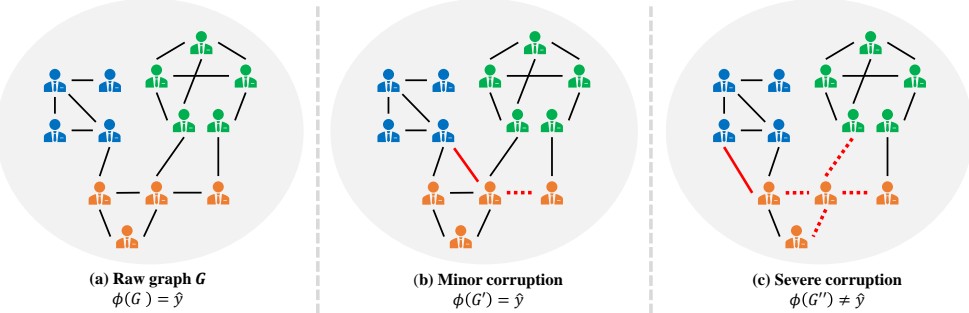

**(a) Raw graph $G$**
$\phi(G) = \hat{y}$

**(b) Minor corruption**
$\phi(G') = \hat{y}$

**(c) Severe corruption**
$\phi(G'') \neq \hat{y}$

Figure 1: An illustration of the structurally corrupted graphs, where the solid red lines represent the newly added edges and the dashed red lines represent the removed edges. (a) is the raw graph and $\hat{y}$ is the prediction given by GNN $\phi$; (b) represents the graph $G'$ with minor corruption whose prediction remains $\hat{y}$; (c) shows the graph $G''$ is severely corrupted such that the predicted label of GNN on it is changed.

GNN explainer aims to identify an explanatory subgraph, whose components exert vital influence on the prediction of the full graph [13]. Existing GNN explanation methods generally allocate significance scores for each component of the graph, and a subgraph with the highest score is extracted as the explanation of the whole graph [14]. They optimize the allocation of the significance scores by minimizing an explanation loss function [15, 13]. The final explanation provided by GNN explainer includes a subgraph structure along with the corresponding node features.

However, a major limitation of existing works is that they mostly focus on designing effective significance score allocation methods, but ignore such a critical problem: *Whether and to what extent a structurally corrupted graph will affect the performance of the GNN explainer?* In many real application scenarios, the graphs may contain various corruptions, such as noise [16, 17] and adversarial edges [18, 19, 20]. Existing methods directly adopt the raw graph features [21, 22, 15, 23] or the latent graph representations [13, 24, 25] as the input of the explainer model, and thus the output explanation may not be reliable when the input is corrupted. Previous work has shown that the explanation of deep neural networks is fragile to attacked images [26]. Thus whether the structurally corrupted graphs will remarkably affect the performance of the GNN explainer and how to address it is an interesting problem, yet has not been fully studied and remains an open research issue.

This paper for the first time investigates how to construct a robust GNN explainer for structurally corrupted graphs. As illustrated in Figure 1, based on whether the prediction of the downstream task (e.g., graph classification) is changed, the structural corruptions of a graph can be categorized into minor corruptions and severe corruptions. In Figure 1(a), $\phi$ denotes the GNN model to be explained, and $G$ is the raw graph. Figure 1(b) shows the case of the graph with minor corruption $G'$, which is not so intense that the predicted graph label remains the same as the raw graph. Figure 1(c) shows the case of the severely corrupted graph $G''$ where the corruption is devastating enough to the GNN and thus results in a different prediction. Based on this example, we argue that a robust GNN explainer should satisfy the following two requirements. First, the minor corruption should be considered as redundant components and filtered by the explainer as they do not affect the final prediction. Second, the severe corruption should be identified by the explainer as part of the explanation since they are the non-ignorable reason that changes the final prediction. However, it is non-trivial to meet the two requirements simultaneously due to the following two challenges.

**Robust graph representation extraction to eliminate the minor corruptions**. As we discussed before, existing works directly use the raw graph features or the latent graph representations of each single graph as the explanation input, which will raise the risk of the GNN explainer overfitting to corruptions [26]. This is because the raw graph features can be easily corrupted by noise or adversarial attacked edges. The latent graph representations have been proved vulnerable to structural corruption [27, 20]. How to extract a robust graph representation that can effectively filter the minor corruptions while preserving the vital information for GNN explanation exploration is challenging.

**Structural irregularity of the severe corruptions**. The corruptions may be of different sizes or consist of several disconnected parts in the graphs, i.e., they are irregular [28]. For example, in molecule chemistry, the corruptive functional groups differ largely in terms of size (e.g., -Cl and

-$C_{10}H_9$ in solubility analysis). Moreover, a molecule containing multiple functional groups which are disconnected from each other is common in the real-world. To meet the second requirement, the structural irregularity of severe corruption should be identified by the GNN explainer. Hence it is not reasonable to adopt some predefined rigorous constraints, e.g., size constraint or connectivity constraint that are commonly used in previous explainers, to the explanations of corrupted graphs. How to design a more general objective function to adaptively capture the structural irregularity of severe corruption is also challenging.

To address the two aforementioned challenges, we propose a robust GNN explainer for structurally corrupted graphs called V-InfoR (Variational Graph Representation based Information bottleneck Robust explainer). Specifically, V-InfoR contains two major modules. (1) **Robust graph representations extractor**. In this module, we take insights of variational inference [29, 30] to model the statistical characteristics that are shared by all the observed graphs, i.e., the mean and the standard deviation. The statistics are capable of capturing the uncertainty and variability of the observed graphs. Based on the statistics, a variational graph distribution is induced, which can effectively model the common features of all the graphs and filter the minor corruption. The variational graph representations provide a more robust graph representation for the downstream explanation generator. (2) **Adaptive explanation generator**. To address the irregularity issue of the severe corruptions, we propose an Adaptive Graph Information (AGI) constraint. The AGI constraint directly restricts the information carried by the explanation without any rigorous assumptions on the structural properties (e.g., size or connectivity) of the explanation. By incorporating the AGI constraint, we formulate the GNN explanation exploration as a Graph Information Bottleneck (GIB) optimization problem to adaptively capture both the regularity and irregularity of the severely corrupted graphs.

Our main contributions are summarized as follows:

- This work for the first time studies the negative effect of both minor and severe structural corruptions on existing GNN explainers, and proposes a robust explainer V-InfoR to effectively handle the two types of corruption.

- We novelly propose to incorporate variational inference to explore GNN explanation. A variational inference based robust graph information extractor is proposed to mitigate the uncertainty and variability of minor corruptions when extracting the critical graph information.

- We generalize the GNN explanation exploration by introducing an adaptive graph information constraint, which can capture both the regularity and irregularity of the corrupted graphs. We also theoretically derive the variational bound of this objective to make it feasibly optimizable.

- Extensive experiments on one synthetic dataset and three real-world datasets demonstrate the superiority of V-InfoR over state-of-the-art GNN explanation methods.

## 2 Preliminary and Problem Statement

In this section, we will first provide the quantitative evidence that existing GNNs explainers are fragile to structurally corrupted graphs. Then we provide a formal statement of the graph explanation problem. See Appendix B for the basic notations.

### 2.1 Are existing GNN explainers robust to corruptions?

To study whether existing GNN explainers are robust to corruptions, we evaluate the performance of six state-of-the-art GNN explainers when minor corruption and severe corruption are injected into the raw graphs, respectively. We use the synthetic BA-3Motifs dataset [15], which is a widely used benchmark dataset for GNN explanation. To investigate the minor corruption, we select the corrupted graphs whose predictions are unchanged after randomly flipping 20% edges. For the severe corruption, we use the SOTA graph adversarial attack algorithm GRABNEL, which formulates the attack task as a Bayesian optimization problem [18]. The attack budget of GRABNEL means the highest percentage of the attacked edges. We use a 10% attack budget here, which is able to change the prediction of the selected graphs and thus can be considered as the severe corruption. $F_{NS}$ score is used as the evaluation metric and a higher $F_{NS}$ score means a better explanation performance [25]. The formal definition will be given later in Section 4.1.

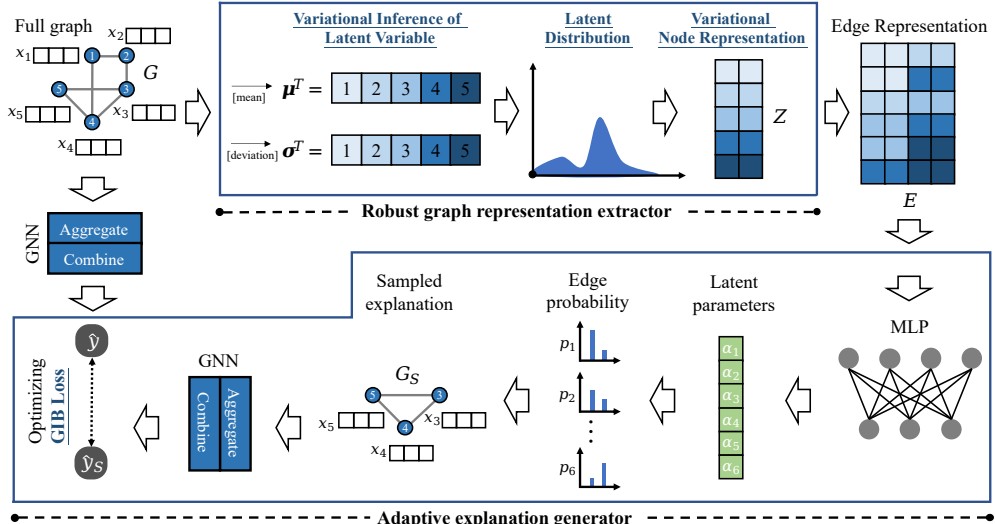

Figure 2: The overall architecture of V-InfoR, which consists of the robust graph representation extractor and the adapative explanation generator.

The result is presented in Figure 5 of Appendix C. The histograms of the three colors represent the explainer performance on raw graphs, graphs with random noise, and attacked graphs, respectively. One can see that both the minor corruption (20% random structural noise) and the severe corruption (10% GRABNEL attack) remarkably degrade the explanation performance. Minor corruption brings about an average performance degradation of 31.15% for the six GNN explainers, and severe corruption results in an average performance degradation of 40.64%. *The experimental result verifies that both the minor and the severe corruptions do significantly affect the performance of existing GNN explainers.* Therefore, a robust GNN explainer is required for the structurally corrupted graphs.

## 2.2 Problem formulation

Formally, given the GNN model $\phi$ to be explained and the input graph $G = (\mathbf{X}, \mathbf{A})$, the explanation provided by the GNN explainer is a subgraph $G_S = (\mathbf{X_S}, \mathbf{A_S})$. $\mathbf{A_S}$ is a subset of the original adjacent matrix $\mathbf{A}$ and contains the key structure that determines prediction largely and $\mathbf{X_S}$ is the features of nodes which associates with $\mathbf{A_S}$. Existing researches show that topological information is critically important to graph classification [27, 13]. Hence, our V-InfoR focuses on the structural corruption when exploring explanation. We deem an explanation $G_S$ *sufficient* when it can produce the same prediction as using the full graph, and deem it *necessary* when the prediction will change if it is removed from the full graph [25]. An ideal explanation should be both sufficient and necessary. Formally, the sufficient and necessary conditions [25] are formulated as follows, respectively,

$$\underset{c \in \mathcal{C}}{\arg\max} \, P_\phi(c|\mathbf{X_S}, \mathbf{A_S}) = \hat{y}. \tag{1}$$

$$\underset{c \in \mathcal{C}}{\arg\max} \, P_\phi(c|\mathbf{X} - \mathbf{X_S}, \mathbf{A} - \mathbf{A_S}) \neq \hat{y}, \tag{2}$$

where $\hat{y} = \arg\max_{c \in \mathcal{C}} P_\phi(c|\mathbf{X}, \mathbf{A})$ in Formula (1) and (2).

## 3 Methodology

Figure 2 illustrates the overall architecture of V-InfoR, which is composed of two key modules, the robust graph representation extractor and the adaptive explanation generator. As shown in the upper part of Figure 2, the robust graph representation extractor takes the full graph $G$ as input, and two GCN encoders are adopted to infer the statistics of $G$'s latent distribution (i.e., mean and standard deviation). The variational node representations, which are sampled from the latent graph distribution, will induce the edge representations. The edge representations are then fed into the adaptive explanation generator which is shown in the lower part of the figure. A multi-layer perceptron

projects the edge representations into the probability of each edge. The explanation subgraph $G_S$ is generated on the basis of this probability. Finally, the proposed novel Graph Information Bottleneck (GIB) objective is optimized, which is defined over the full graph prediction $\hat{y}$ and the explanation subgraph prediction $\hat{y}_S$. Next, we will introduce the two modules in detail.

## 3.1  Robust graph representation extractor

Specifically, we introduce a variational auto-encoder (VAE) [30] to deduce the statistics in the variational distribution of the minorly corrupted graphs. For a set of observed graphs $\mathcal{G} = \{G^k = (\mathbf{X}^k, \mathbf{A}^k)\}_{k=1}^{|\mathcal{G}|}$, variational inference assumes that they are generated by a latent random process, involving a latent continuous random variable $\mathbf{z}$ [29, 30]. This random process contains two steps. First, a value $\mathbf{z}^k$ is generated from a prior distribution $p(\mathbf{z})$. Then the graph $G^k$ is generated from a conditional distribution $p(\mathbf{G}|\mathbf{z})$ and observed by us. Variational inference aims at identifying the posterior distribution $p(\mathbf{z}|\mathbf{G})$, i.e., inverting the distribution of latent variable $\mathbf{z}$ from the observed graphs [4]. However, the true posterior distribution $p(\mathbf{z}|\mathbf{G})$ is unknown. A feasible method is introducing a variational distribution $q(\mathbf{z}; \mathbf{G})$ to approximate $p(\mathbf{z}|\mathbf{G})$ [28]. This work assumes that $q(\mathbf{z}; \mathbf{G})$ is a Gaussian distribution $\mathcal{N}(\boldsymbol{\mu}, \boldsymbol{\sigma}^2)$ and increases the similarity between $q(\mathbf{z}; \mathbf{G})$ and $p(\mathbf{z}|\mathbf{G})$. The similarity measurement is Kullback-Leibler divergence,

$$\mathrm{KL}[q(\mathbf{z}; \mathbf{G}) || p(\mathbf{z}|\mathbf{G})] = \sum_{\mathbf{z}^k} q(\mathbf{z}^k; \mathbf{G}) \log \frac{q(\mathbf{z}^k; \mathbf{G})}{p(\mathbf{z}^k|\mathbf{G})}. \tag{3}$$

We substitute the unknown $p(\mathbf{z}|\mathbf{G})$ for $p(\mathbf{z}, \mathbf{G})/p(\mathbf{G})$. Finally, the objective function of the robust graph representation extractor to be minimized can be formulated as

$$L_{\mathrm{VAE}} = -\mathbb{E}_{q(\mathbf{z}; \mathbf{G})}[\log p(\mathbf{G}|\mathbf{z})] + \mathrm{KL}[q(\mathbf{z}; \mathbf{G}) || p(\mathbf{z})], \tag{4}$$

where $p(\mathbf{z})$ is the standard Gaussian distribution $\mathcal{N}(\mathbf{0}, \mathbf{I})$. See Appendix A for detailed derivation.

We refer to the latent variable $\mathbf{z}$ as a variational representation because it contains the indicative information of the observed graphs. We also refer to the model that computes the distribution $q(\mathbf{z}; \mathbf{G})$ as an *encoder*, since given a graph $G$, it produces a distribution over the latent variable $\mathbf{z}$, from which $G$ could have been generated. Similarly, $p(\mathbf{G}|\mathbf{z})$ is referred to as a *decoder*, since given a code $\mathbf{z}$ it produces the probability of the possible graph $G$.

In our implementation, we employ a 3-layer GCN as the encoder and a simple inner product as the decoder. The first two layers of GCN aim to aggregate and combine the information from the graph. The third layer consists of two separate graph convolution operators to calculate the mean $\boldsymbol{\mu}$ and standard deviation $\boldsymbol{\sigma}$ of $q(\mathbf{z}; \mathbf{G})$, respectively. This procedure can be formulated as

$$\mathbf{H}^k = \mathrm{GCN}(\mathbf{H}^{k-1}, \mathbf{A}), k = 1, 2, \mathbf{H}^0 = \mathbf{X}, \tag{5}$$

$$\boldsymbol{\mu} = \mathrm{GCN}_{\boldsymbol{\mu}}(\mathbf{H}^2, \mathbf{A}), \tag{6}$$

$$\log \boldsymbol{\sigma} = \mathrm{GCN}_{\boldsymbol{\sigma}}(\mathbf{H}^2, \mathbf{A}). \tag{7}$$

The variational inference procedure endows $q(\mathbf{z}; \mathbf{G})$ vital graph information and is insensitive to minor corruptions. Hence the sampled graph representation $\mathbf{z}$ which follows $q(\mathbf{z}; \mathbf{G})$ is more robust. Next, the robust representation $\mathbf{z}$ will be fed into the explanation generator for explanation exploration.

## 3.2  Adaptive explanation generator

As the second challenge stated before, severe corruptions should be identified as part of the explanation, but they are usually irregular in terms of size, connectivity or some other structural properties. The rigorous constraints of the structural regularity adopted in previous works [15, 13, 25] are thus not feasible for the severe corruptions. Hence, a new GNN explanation exploration objective function is required to adaptively capture the irregularity of the severe corruptions. Inspired by the Graph Information Bottleneck (GIB) principle [31, 32], we propose to introduce an Adaptive Graph Information (AGI) constraint in exploring GNN explanation. On the one hand, the AGI constraint functions as a bottleneck to directly restrict the information carried by the explanation $G_S$, instead

---
[4]Without ambiguity, $\mathbf{z}$ represents the random variable and its sampled value (i.e., $\mathbf{z}^k$), simultaneously.

of simply restricting the size or connectivity of $G_S$. Without any predefined structural regularity constraints, our method can more effectively capture the irregularity of the explanation caused by the severe corruptions. On the other hand, since the information measurement of the explanation $G_S$ can be continuously optimized, the AGI constraint can adaptively cover the discrete rigorous constraints [28]. To this end, we formulate the GNN explanation problem as a GIB-guided optimization task to adaptively generate the explanations.

The insight of the GIB-guided optimization is that the explanation $G_S$ of graph $G$ should contain the *minimal sufficient* components of $G$ [31]. GIB principle facilitates $G_S$ to be informative enough about the prediction $\hat{y}$ of $G$ (*sufficient*). GIB principle also inhibits $G_S$ from preserving redundant components which is irrelevant for predicting $G$ (*minimal*). To this aim, the GIB-guided optimization task of $G_S$ is formally defined as follows,

$$\min_{G_S \subset G} \text{GIB}(G, \hat{y}; G_S) = -\text{MI}(\hat{y}, G_S) + \beta \text{MI}(G, G_S), \tag{8}$$

where $\text{MI}(\cdot, \cdot)$ denotes the mutual information. The second term $\text{MI}(G, G_S)$ measures the vital graph information carried by the explanation $G_S$, which functions as the AGI constraint. Nevertheless, the GIB-guided explanation exploration task in Formula (8) cannot be directly extended to a continuous optimization procedure, because both $G$ and $G_S$ have discrete topological information which is difficult to optimize over. We resort to Gilbert random graph theory [33] which argues that an arbitrary graph $G$ can be represented as a random graph variable, and each edge of $G$ is associated with a binary random variable $r$ to reveal its existence. Additionally, the existence of one edge is conditionally independent of the other edges. $r_{ij} = 1$ means there is an edge $(i, j)$ from $v_i$ to $v_j$, otherwise $r_{ij} = 0$. To sum up, an arbitrary graph $G$ can be represented as

$$p(G) = \prod_{(i,j)} p(r_{ij}). \tag{9}$$

For the binary variable $r_{ij}$, a common instantiation is the Bernoulli distribution $r_{ij} \sim \text{Bern}(\theta_{ij})$, where $\theta_{ij} = p(r_{ij} = 1)$ is the probability of edge $(i, j)$ existing in $G$. However, the Bernoulli distributiont cannot be directly optimized. To address this issue, we apply categorical reparameterization [34] to the Bernoulli variable $r_{ij}$. The continuous relaxation of $r_{ij}$ can be formulated as

$$r_{ij} = \text{Sigmoid}\left(\frac{\log \epsilon - \log(1 - \epsilon) + \alpha_{ij}}{\tau}\right), \epsilon \sim \text{Uniform}(0, 1), \tag{10}$$

where we let the latent parameter $\alpha_{ij} = \log \frac{\theta_{ij}}{1 - \theta_{ij}}$. $\tau$ controls the approximation between the relaxed distribution and $\text{Bern}(\theta_{ij})$. When $\tau$ approaches 0, the limitation of Formula (10) is $\text{Bern}(\theta_{ij})$.

According to Formula (10), the Bernoulli parameter $\theta_{ij}$ is indeed associated with parameter $\alpha_{ij}$. In our implementation, we use a multi-layer perceptron (MLP) to compute $\boldsymbol{\alpha}$. The MLP takes the variational node representation $\mathbf{z}$ as input and concatenates the representations of two nodes $v_i, v_j$ as the representation of the corresponding edge $(i, j)$, which can be formulated as

$$\alpha_{ij} = \text{MLP}[(\mathbf{z}_i, \mathbf{z}_j)], \tag{11}$$

where $[\cdot, \cdot]$ is the concatenation operator.

Based on $\boldsymbol{\alpha}$ and Formula (10), we obtain the probability matrix $\mathbf{M_p}$ whose elements indicate the existence of the corresponding edges. Next, we can sample the explanation $G_S$ based on the probabilities in the matrix $\mathbf{M_p}$ as follows,

$$G_S = (\mathbf{X_S}, \mathbf{A_S} = \mathbf{M_p} \odot \mathbf{A}). \tag{12}$$

So far, we have derived the optimizable representation of $G_S$ in Formula (8). However, the optimization is still challenging since the distributions $p(\hat{y}|G_S)$ and $p(G_S)$ are intractable.

Fortunately, following the variational approximation proposed in [35], we can derive a tractable variational upper bound of GIB in Formula (8). For the first term $-\text{MI}(\hat{y}, G_S)$, a parameterized variational approximation $p_\phi(\hat{y}|G_S)$ is introduced for $p(\hat{y}|G_S)$ to get the upper bound as follows,

$$-\text{MI}(\hat{y}, G_S) \leq -\mathbb{E}_{p(G_S, \hat{y})}\left[\log p_\phi(\hat{y}|G_S)\right] + H(\hat{y}), \tag{13}$$

where $p_\phi(\hat{y}|G_S)$ is the GNN model and $H(\hat{y})$ is an entropy independent of $G_S$. For the second term, we introduce $q(G_S)$ for the marginal distribution $P(G_S)$ [28], and the upper bound is

$$\text{MI}(G_S, G) \leq \mathbb{E}_{p(G)}\left[\text{KL}(p_\alpha(G_S|G)||q(G_S))\right], \tag{14}$$

where $p_\alpha(G_S|G)$ represents the explanation generator and $q(G_S)$ represents the prior distribution sharing a similar spirit of assuming standard Gaussian prior [28]. See Appendix A for the detailed derivation. Finally, we obtain the variational upper bound of the GIB Formula (8) as follows,

$$L_{\text{GIB}} = - \mathbb{E}_{p(G_S, \hat{y})} \big[ \log p_\phi(\hat{y}|G_S) \big] + \beta \mathbb{E}_{p(G)} \big[ \text{KL}(p_\alpha(G_S|G) || q(G_S)) \big], \tag{15}$$

where $q(G_S)$ is usually set as

$$q(G_S) = C \cdot \prod_{i,j=1}^{N} p_\pi(e_{ij}), e_{ij} \sim \text{Bern}(\pi), \tag{16}$$

where $C$ is a constant that decided by the hyper-parameter $\pi$.

Note that there are no rigorous constraints imposed in Formula (15). The second term actually functions as an adaptive constraint that inhibits $G_S$ from containing useless information. The risk of ignoring the irregular severe corruptions in the final explanation $G_S$ can also be largely mitigated.

To achieve a robust GNN explainer for all the corruption situations, the two proposed modules are trained jointly by minimizing the following overall objective function,

$$L_{\text{joint}} = L_{\text{VAE}} + L_{\text{GIB}}. \tag{17}$$

## 4 Experiment

### 4.1 Experimental setup

Following the standard procedure of GNN explanation [15, 2], we conduct two steps in the experiments: (1) training a base GNN model for prediction, and (2) generating explanations for this base model. For the first step, the base GNN model is trained to simulate the third-party predictor which is independent of the GNN explainers. We take the trained GNN model as an oracle during the explanation generation, i.e., input the graph (no matter clean or corrupted) and get the prediction. In the second step, we simulate the corruptions by introducing random noise or adversarial attack to the graph structures. The random noise represents the corruptions that naturally exist in real-world which affects each graph component without any distinction. The noise ratio controls the percentage of randomly flipped edges. The adversarial attack represents the man-made malicious corruptions and the attack budget is the highest percentage of attacked edges.

**Metric.** For a quantitative explanation evaluation, we report the probability of sufficient $P_S$, the probability of necessary $P_N$ and the $F_{NS}$ scores. See Appendix D.1 for the detailed definition.

**Dataset.** We evaluate the proposed V-InfoR and baseline explanation methods on one synthetic dataset and three real-world datasets. The synthetic dataset is BA-3Motifs introduced in [36]. Three real-world datasets are Mutag, Ogbg-molhiv and Ogbg-ppa.

**Baseline.** The comparable baseline explainers include gradients-based methods GradCAM [21] and IG [22], surrogate method PGM-Explainer [37], and perturbation-based methods GNNExplainer [15], PGExplainer [2] and ReFine [36].

The detailed descriptions of datasets, baselines, and base GNN models are given in Appendix D. The ablation study is presented in Appendix E. We also report the visualized cases of GNN explanation for qualitative analysis in Appendix F.

### 4.2 Quantitative Evaluation on Graphs with Random Noise

We report the result on the graphs with 20% random structural noise corruptions in Table 1. Specifically, we randomly select the edges according to the noise ratio and flip the selected ones.

The result shows that V-InfoR is able to improve the explainer performance on the four datasets, with the overall metric $F_{NS}$ score improvement at least by 9.19% for the BA-3Motifs dataset and the highest improvement by 29.23% for the Mutag dataset. The V-InfoR improves the performance remarkably for Mutag by 29.23% and Ogbg-molhiv datasets by 15.65%, since the irregularity of the corruptions in chemical molecule structure is more obvious than the other types of graphs (motifs in BA-3Motifs and biologically associations in Ogbg-ppa), and V-InfoR is able to effectively capture

Table 1: The comparison of V-InfoR and baselines under random structural noise. We use bold font to mark the highest score. The second highest score is marked with underline. The Impro. is defined as $\big([\text{V-InfoR}]\text{-}[\text{Best Baseline}]\big)\big/[\text{Best Baseline}]$.

| Dataset | Metric | GradCAM | IG | GNNExplainer | PGExplainer | PGM-Explainer | ReFine | V-InfoR | Rank | Impro. |
|---|---|---|---|---|---|---|---|---|---|---|
| BA-3Motifs | $P_S$ | 0.8725 | 0.8625 | 0.8535 | 0.8510 | 0.8505 | 0.8300 | **0.8820** | 1 | 1.09% |
| | $P_N$ | 0.2605 | 0.2795 | 0.2410 | 0.2095 | 0.2235 | 0.2625 | **0.3021** | 1 | 8.09% |
| | $F_{NS}$ | 0.4012 | 0.4222 | 0.3758 | 0.3362 | 0.3540 | 0.3989 | **0.4610** | 1 | 9.19% |
| Mutag | $P_S$ | 0.8760 | 0.8880 | 0.8916 | 0.8640 | 0.8900 | 0.8900 | **0.8964** | 1 | 0.54% |
| | $P_N$ | 0.0996 | 0.1068 | 0.1080 | 0.1260 | 0.1020 | 0.1260 | **0.1696** | 1 | 34.60% |
| | $F_{NS}$ | 0.1789 | 0.1907 | 0.1920 | 0.2199 | 0.1830 | 0.2207 | **0.2852** | 1 | 29.23% |
| Ogbg-molhiv | $P_S$ | 0.9230 | 0.9200 | 0.8925 | 0.8390 | 0.8860 | 0.9105 | **0.9386** | 1 | 1.69% |
| | $P_N$ | 0.0680 | 0.0400 | 0.0940 | 0.1265 | 0.0980 | 0.1020 | **0.1470** | 1 | 16.21% |
| | $F_{NS}$ | 0.1267 | 0.0767 | 0.1701 | 0.2198 | 0.1765 | 0.1834 | **0.2542** | 1 | 15.65% |
| Ogbg-ppa | $P_S$ | 0.4340 | 0.5820 | 0.6616 | 0.6260 | 0.6192 | 0.6344 | **0.6700** | 1 | 1.27% |
| | $P_N$ | 0.4720 | 0.4600 | 0.3480 | 0.2856 | 0.3780 | 0.4406 | **0.4930** | 1 | 4.45% |
| | $F_{NS}$ | 0.4522 | 0.5139 | 0.4561 | 0.3922 | 0.4694 | 0.5200 | **0.5680** | 1 | 9.23% |

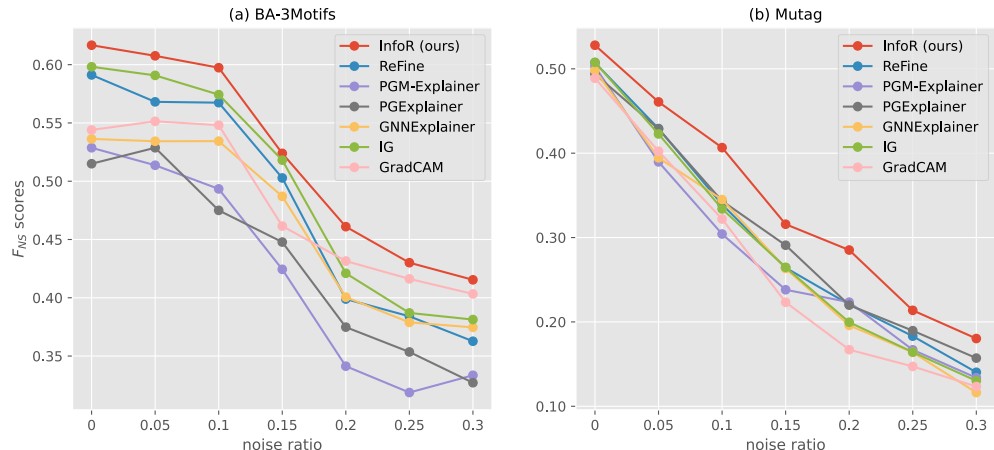

Figure 3: The comparison of V-InfoR and six baselines under different noise ratios in (a) BA-3Motifs and (b) Mutag.

the irregularity. For the BA-3Motifs and Ogbg-ppa datasets with less obvious irregularity, V-InfoR can still achieve a performance improvement, since the information constraint is general enough to cover the rigorous constraint and thus avoid performance degradation.

One can also see that the raw features based explainers, including GradCAM, IG, GNNExplainer, and PGM-Explainer, perform poorly in small/medium-scale datasets (BA-3Mmotifs, Mutag and Ogbg-molhiv). This may be because the raw features are more easily to be corrupted in small/medium-scale graphs. Since the graph representation is more difficult to learn in the large-scale graph, PGExplainer and ReFine that rely on latent graph representations are unable to well explain large-scale graphs for the Ogbg-ppa dataset. V-InfoR adopts the robust graph representation which is sampled from a variational distribution containing common information shared by large amounts of graphs, and thus achieves an overall better performance. Furthermore, one can note that all baseline GNN explainers achieve a high $P_S$, while the $P_N$ is relatively low. This phenomenon implies that the sufficient condition is easier to satisfy than the necessary condition.

As shown in Figure 3, we also evaluate V-InfoR and baselines under different noise ratios ranging from 0 to 0.30, which reverals the tendency of explainer performance with the increase of noise ratio. In real application scenarios, it is intractable to separate the minor and the severe corruptions, and both may exist simultaneously. Different noise ratios indicate different mixing ratios of the two corruptions. Note that $F_{NS}$ scores with zero noise ratio represent the result on raw graphs without any corruption. It shows that V-InfoR still achieves the best performance when no noise is added (noise ratio = 0). This reveals that even without any corruption, the robust representation extractor still extracts vital common graph information and the explanation generator adaptively identifies the explanations with regular structural properties.

Table 2: The comparison of V-InfoR and baselines under GRABNEL attack [18]. We use bold font to mark the highest score. The second highest score is marked with underlines.

| Attack budegt | Dataset | Metric | GradCAM | IG | GNNExplainer | PGExplainer | PGM-Explainer | ReFine | V-InfoR | Rank | Impro. |
|---|---|---|---|---|---|---|---|---|---|---|---|
| 5% | BA-3Motifs | $P_S$ | 0.6980 | 0.6925 | 0.5625 | 0.6225 | 0.5950 | 0.6700 | **0.7075** | 1 | 1.36% |
| | | $P_N$ | 0.3625 | 0.4675 | 0.4200 | 0.3700 | 0.3925 | 0.3925 | **0.5450** | 1 | 16.58% |
| | | $F_{NS}$ | 0.4772 | 0.5582 | 0.4809 | 0.4641 | 0.4730 | 0.4950 | **0.6157** | 1 | 10.30% |
| | Mutag | $P_S$ | 0.5740 | 0.6600 | 0.6140 | 0.6610 | 0.5820 | 0.6340 | **0.6760** | 1 | 2.27% |
| | | $P_N$ | 0.4200 | 0.3875 | 0.3800 | 0.4003 | 0.4060 | 0.4100 | **0.4588** | 1 | 9.24% |
| | | $F_{NS}$ | 0.4851 | 0.4883 | 0.4695 | 0.4986 | 0.4783 | 0.4980 | **0.5466** | 1 | 9.63% |
| 10% | BA-3Motifs | $P_S$ | 0.8720 | 0.8495 | 0.8605 | 0.9020 | 0.8125 | 0.8800 | **0.9185** | 1 | 1.83% |
| | | $P_N$ | 0.0800 | 0.2105 | 0.2615 | 0.2815 | 0.1925 | 0.2100 | **0.3332** | 1 | 18.37% |
| | | $F_{NS}$ | 0.1466 | 0.3374 | 0.4011 | 0.4291 | 0.3113 | 0.3391 | **0.4890** | 1 | 13.96% |
| | Mutag | $P_S$ | 0.5848 | 0.7370 | 0.6616 | 0.6524 | 0.6392 | 0.6140 | **0.7424** | 1 | 0.73% |
| | | $P_N$ | 0.4160 | 0.3404 | 0.3284 | 0.3928 | 0.3344 | 0.4040 | **0.4277** | 1 | 2.81% |
| | | $F_{NS}$ | 0.4862 | 0.4657 | 0.4389 | 0.4904 | 0.4391 | 0.4873 | **0.5427** | 1 | 10.66% |

## 4.3 Quantitative Evaluation on Graphs with Attack Corruption

The adversarial attack corresponds to the malicious corruption painstakingly customized by the attacker. Here we use the SOTA graph classification attack algorithm GRABNEL [18], whose attack budget is the highest percentage of attacked edges. GRABNEL employs a surrogate model to learn the mapping from an attacked graph to its attack loss and optimizes the attacked graph iteratively via an adapted genetic algorithm until successful attack or budget exhaustion. Considering the overhead of executing GRABNEL, we run the attack algorithm only on the BA-3Motifs and Mutag datasets. We report the results with 5% and 10% attack budgets in Table 2.

The result reveals that InfoR achieves an overall superior performance over the other baselines in terms of all metrics. As the attack budget grows from 5% to 10%, though the GNN explanation task becomes harder (all metrics decrease), the performance improvement of V-InfoR becomes more significant than baselines. This shows that V-InfoR can effectively mitigate the negative influence of the structural corruptions. For the Mutag dataset, the performance improvement on the graphs with GRABNEL attack (13.96%) is less significant than that on the graphs with random noise (29.23%). This gap can be attributed to the difference between adversarial attack and random noise. Although the adversarial attack corruption is devastating to change the model prediction, they are subtle and introduce as less irregularity as possible to the raw graphs.

## 4.4 Hyper-parameter analysis

We further analyse the effect of three parameters on the model performance, $\tau$, $\beta$ and $\pi$. $\tau$ controls the approximation degree of $r_{ij}$ distribution to Bernoulli distribution, which ranges in $[0.1, 0.5]$. $\beta$ controls the balance between the strength of information restoring (i.e., $\min -\text{MI}(\hat{y}, G_S)$) and the strength of information filtering (i.e., $\min \text{MI}(G_S, G)$). $\pi$ represents the prior Bernoulli probability, which controls the distribution of $q(G_S)$.

Figure 4 shows the influence of the three hyper-parameters on V-InfoR for the BA-3Motifs and Mutag datasets. One can roughly archieve the following three conclusions. First, V-InfoR is not so sensitive to $\beta$ that controls the strength of AGI constraint, which verifies the adaptability of our proposed constraint. Second, a suitable value of $\tau$ is around 0.3, which means the best balance between the continuity of Formula (10) and the approximation degree is achieved when $\tau = 0.3$. Third, there is no obvious pattern shown in Figure 4(c) for the choice of $\pi$ in Formula (16), which means that a reasonable $\pi$ may largely depend on the specific dataset.

## 5 Related Work

Early attempts to explain GNN simply transfer gradients-based methods to graph data, and they regard the gradients of nodes and edges as significance scores [14]. Though efficient and intuitive, the explanations based on gradients are unavoidably suffered from gradients saturation [25]. Surrogate methods [37, 23] are also adopted in GNN explanations. Yet limited by simple models that function as surrogates, these approaches are unable to capture topological structure which plays an important role in GNN predictions [13, 27]. Currently, the most concerned GNN explanation methods are perturbation-based. By intervening components in the graph, such as deleting (or adding) edges

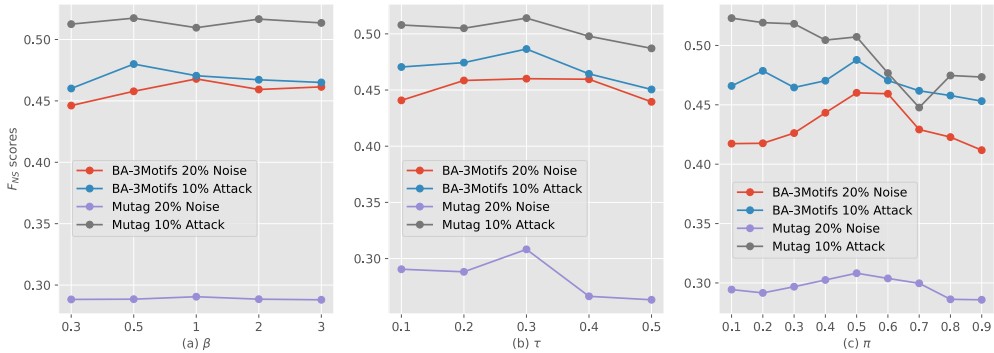

Figure 4: The influence of hyper-parameters $\beta$ in Formula (8), $\tau$ in Formula (10) and $\pi$ in Formula (16) on V-InfoR.

(or nodes), and monitoring the change of the corresponding prediction, perturbation-based GNN explanation methods [15, 13, 24, 38] optimize the significance scores matrix round after round.

As a classic perturbation-based GNN explanation method, GNNExplainer [15] determines the significant components by maximizing the mutual information between the intervened and the original graphs. It calculates significance scores for both node features and edges, and the components whose scores are below-threshold will be removed. PGExplainer [13] introduces a parameterized neural network to provide significance scores. It demands a training procedure to endow the internal neural network with multi-categorical predicting behavior, and the trained PGExplainer can explain any new graph without retraining. While the previous methods aim to preserve the components that make the prediction invariant, CF-GNNExplainer aims to find the components that will make the prediction change if they are missing [24]. XGNN [39] formulates the GNN explanation problem as a reinforcement learning task. Starting with an empty graph, XGNN gradually adds components until the generated graph belongs to the specified class.

## 6  Conclusion

In this paper, we propose a robust GNN explainer V-InfoR for the structurally corrupted graphs. V-InfoR employs the variational inference to learn the robust graph representations and generalizes the GNN explanation exploration to a graph information bottleneck (GIB) optimization task without any predefined rigorous constraints. The robust graph representations are insensitive to minor corruptions when extract the common information shared by the observed graphs. By introducing an adaptive graph information constraint, V-InfoR can effectively capture both the regularity and irregularity of the explanation subgraphs. Extensive experiments demonstrate its superior performance and the ablation study further illustrates the effectiveness of two proposed modules.

## Acknowledgement

This research was funded by the National Science Foundation of China (No. 62172443), Open Project of Xiangjiang Laboratory (22XJ03010, 22XJ03005), the Science and Technology Major Project of Changsha (No. kh2202004), Hunan Provincial Natural Science Foundation of China (No. 2022JJ30053), and the High Performance Computing Center of Central South University.

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

# A Detailed Derivation

First, we give the detailed derivation of Formula (4). To maximize the similarity between the variational distribution and the true posterior distribution, we minimize the Kullback-Leibler divergence,

$$
\begin{aligned}
\mathrm{KL}[q(\mathbf{z}; \mathbf{G}) &\| p(\mathbf{z_i}|\mathbf{G})] \\
&= \sum_{\mathbf{z_i}} q(\mathbf{z_i}; \mathbf{G}) \log \frac{q(\mathbf{z_i}; \mathbf{G})}{p(\mathbf{z_i}|\mathbf{G})} \\
&= \sum_{\mathbf{z_i}} q(\mathbf{z_i}; \mathbf{G}) \log \frac{q(\mathbf{z_i}; \mathbf{G})}{p(\mathbf{z_i}, \mathbf{G})} + \sum_{\mathbf{z_i}} q(\mathbf{z_i}; \mathbf{G}) \log p(\mathbf{G}) \\
&= \sum_{\mathbf{z_i}} q(\mathbf{z_i}; \mathbf{G}) \log \frac{q(\mathbf{z_i}; \mathbf{G})}{p(\mathbf{z_i}, \mathbf{G})} + \log p(\mathbf{G}).
\end{aligned}
\tag{18}
$$

Since $p(\mathbf{G})$ is independent of $\mathbf{z}$, we get the following derivation,

$$
\begin{aligned}
&\sum_{\mathbf{z_i}} q(\mathbf{z_i}; \mathbf{G}) \log \frac{q(\mathbf{z_i}; \mathbf{G})}{p(\mathbf{z_i}, \mathbf{G})} \\
&= \sum_{\mathbf{z_i}} q(\mathbf{z_i}; \mathbf{G}) \log \frac{q(\mathbf{z_i}; \mathbf{G})}{p(\mathbf{G}|\mathbf{z_i}) \cdot p(\mathbf{z_i})} \\
&= \sum_{\mathbf{z_i}} q(\mathbf{z_i}; \mathbf{G}) \log \frac{q(\mathbf{z_i}; \mathbf{G})}{p(\mathbf{z_i})} - \sum_{\mathbf{z_i}} q(\mathbf{z_i}; \mathbf{G}) \log p(\mathbf{G}|\mathbf{z_i}) \\
&= \mathrm{KL}[\mathrm{q}(\mathbf{z}; \mathbf{G}) \| \mathrm{p}(\mathbf{z})] - \mathbb{E}_{q(\mathbf{z}; \mathbf{G})}[\log p(\mathbf{G}|\mathbf{z})].
\end{aligned}
\tag{19}
$$

Next, we deduce the upper bound of $-\mathrm{MI}(\hat{y}, G_S)$ in Formula (13),

$$
\begin{aligned}
-\mathrm{MI}(\hat{y}, G_S) &= -\mathbb{E}_{p(\hat{y}, G_S)}\left[\log \frac{p(\hat{y}, G_S)}{p(\hat{y}) \cdot p(G_S)}\right] \\
&= -\mathbb{E}_{p(\hat{y}, G_S)}\left[\log \frac{p(\hat{y}|G_S)}{p(\hat{y})}\right] \\
&= -\mathbb{E}_{p(\hat{y}, G_S)}\left[\log \frac{p_\phi(\hat{y}|G_S)}{p(\hat{y})}\right] - \mathbb{E}_{p(\hat{y}, G_S)}\left[\log \frac{p(\hat{y}|G_S)}{p_\phi(\hat{y}|G_S)}\right] \\
&= -\mathbb{E}_{p(\hat{y}, G_S)}\left[\log \frac{p_\phi(\hat{y}|G_S)}{p(\hat{y})}\right] - \mathbb{E}_{p(G_S)}\left[\mathrm{KL}\big[p(\hat{y}|G_S) \| p_\phi(\hat{y}, G_S)\big]\right] \\
&\leq -\mathbb{E}_{p(\hat{y}, G_S)}\left[\log \frac{p_\phi(\hat{y}|G_S)}{p(\hat{y})}\right] \\
&= -\mathbb{E}_{p(\hat{y}, G_S)}\big[\log p_\phi(\hat{y}|G_S)\big] + H(\hat{y}).
\end{aligned}
\tag{20}
$$

At last, we deduce the upper bound of $\mathrm{MI}(G_S, G)$ in Formula (14),

$$
\begin{aligned}
\mathrm{MI}(G_S, G) &= \mathbb{E}_{P(G_S, G)}\left[\log \frac{p(G_S, G)}{p(G_S) \cdot p(G)}\right] \\
&= \mathbb{E}_{P(G_S, G)}\left[\log \frac{p_\alpha(G_S|G)}{p(G_S)}\right] \\
&= \mathbb{E}_{P(G_S, G)}\left[\log \frac{p_\alpha(G_S|G)}{q(G_S)}\right] + \mathbb{E}_{P(G_S, G)}\left[\log \frac{q(G_S)}{p(G_S)}\right] \\
&= \mathbb{E}_{P(G_S, G)}\left[\log \frac{p_\alpha(G_S|G)}{q(G_S)}\right] + \mathbb{E}_{P(G|G_S)}\left[p(G_S) \log \frac{q(G_S)}{p(G_S)}\right] \\
&= \mathbb{E}_{P(G_S, G)}\left[\log \frac{p_\alpha(G_S|G)}{q(G_S)}\right] + \mathbb{E}_{P(G|G_S)}\left[-\mathrm{KL}\big[p(G_S) \| q(G_S)\big]\right] \\
&\leq \mathbb{E}_{p(G)}\left[p_\alpha(G_S|G) \log \frac{p_\alpha(G_S|G)}{q(G_S)}\right] \\
&= \mathbb{E}_{p(G)}\left[\mathrm{KL}\big[p_\alpha(G_S|G) \| q(G_S)\big]\right].
\end{aligned}
\tag{21}
$$

# B  Basic Notations

The basic notations and descriptions are summarized in Table 3

Table 3: Basic notations and descriptions in the manuscirpt.

| Notation | Description |
|---|---|
| $G, G^k$ | Graph instance, $k$-th graph instance |
| $G_S$ | Explanatory subgraph |
| $\mathbf{X}, \mathbf{X_S}$ | Node feature matrix |
| $\mathbf{A}, \mathbf{A_S}$ | Adjacency matrix |
| $\mathcal{G}$ | Graph set |
| $\phi$ | GNN prediction model |
| $\hat{y}$ | GNN predicted label |
| $c, \mathcal{C}$ | Specific label, label space |
| $\hat{y}_S$ | GNN prediction label of explanatory subgraph |
| $\lvert \cdot \rvert$ | Cardinality of set |
| $\mathbf{z}, \mathbf{z}^k$ | Continuous random variable, $k$-th sampled value |
| $\mathcal{N}(\boldsymbol{\mu}, \boldsymbol{\sigma}^2)$ | Gaussian distribution with $\boldsymbol{\mu}$ mean and $\boldsymbol{\sigma}$ standard deviation |
| $r_{ij}$ | Bernoulli variable of edge $(i, j)$ |
| $\theta_{ij}$ | Bernoulli parameter of edge $(i, j)$ |
| $\alpha_{ij}$ | Latent parameter of edge $(i, j)$ |
| $\tau$ | Temperature parameter in Concrete distribution |
| $\epsilon$ | Random noise following $\mathcal{N}(0, 1)$ |
| $\mathbf{M_P}$ | Probability matrix |

# C  Preliminary Experiment

The results of six SOTA GNN explainers on the graphs with minor and severe corruptions is shown in Figure 5. The dertailed analysis in reported at Section 2.1.

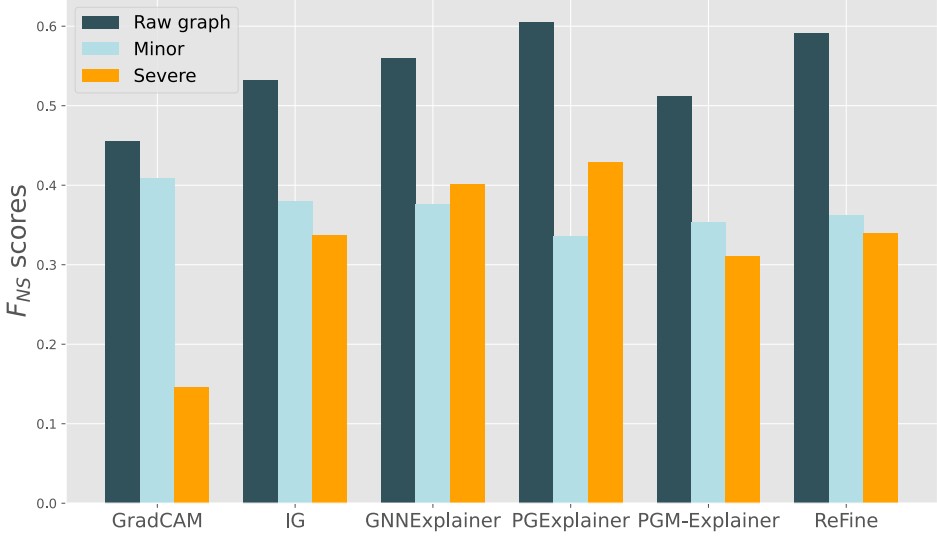

Figure 5: The $F_{NS}$ scores of the six SOTA GNN explainers on BA-3Motifs, BA-3Motifs with minor corruptions, and BA-3Motifs with severe corruptions, respectively.

# D Experiment supplement

## D.1 Metric

We formally define the three metrics for explanation evaluation as follows,

$$P_S = \frac{\sum \mathbb{I}(\tilde{y} = \hat{y})}{|\mathcal{G}_{\text{test}}|}, \text{where } \tilde{y} = \operatorname*{argmax}_{c \in \mathcal{C}} P_\phi(c|\mathbf{X_S}, \mathbf{A_S}), \tag{22}$$

$$P_N = \frac{\sum \mathbb{I}(\tilde{y} \neq \hat{y})}{|\mathcal{G}_{\text{test}}|}, \text{where } \tilde{y} = \operatorname*{argmax}_{c \in \mathcal{C}} P_\phi(c|\mathbf{X} - \mathbf{X_S}, \mathbf{A} - \mathbf{A_S}), \tag{23}$$

$$F_{NS} = \frac{2 \cdot P_S \cdot P_N}{P_S + P_N}, \tag{24}$$

where $\mathcal{G}_{\text{test}}$ is the test dataset. $P_S$ measures how faithfully the explanations in simulating the full graph, $P_N$ measures the indispensability of explanations, and $F_{NS}$ score is the Harmonic mean of $P_S$ and $P_N$, which measures the overall performance of a GNN explanation method.

## D.2 Dataset

We next briefly introduce the four datasets. The statistics of the four datasets are shown in Table 4, where the "*avg nodes/edges*" means the average node/edge number of the graphs and the "*split*" is the dataset splitting of the training, validation, and testing sets.

- **BA-3Motifs** [36] contains 3,000 synthetic graphs. Each graph adopts Barabasi Albert graphs as the base and attaches each base with one of three motifs: house, cycle, and grid. The synthetic graphs are classified into three classes according to the type of attached motifs.

- **Mutag** [6] contains 4,337 molecule graphs which are classified into two categories in accordance with their mutagenic effect on a bacterium.

- **Ogbg-molhiv** [40] contains 41,127 molecule graphs which are classified into two categories based on whether a molecule inhibits HIV virus replication or not.

- **Ogbg-ppa** [40] contains 158,100 protein-protein association networks in which nodes represent proteins and edges indicate biologically meaningful associations. These association networks cover 1,581 species that belong to 37 taxonomic groups (e.g., mammals and bacterial families). The prediction task is a 37-classes classification to predict which taxonomic group the network is from.

Table 4: The statistics of the four datasets.

| Dataset | avg nodes | avg edges | split |
|---------|-----------|-----------|-------|
| BA-3Motifs | 21.92 | 14.76 | 2200/400/400 |
| Mutag | 30.32 | 30.77 | 3337/500/500 |
| Ogbg-molhiv | 25.5 | 27.5 | scaffold split [40] |
| Ogbg-ppa | 243.4 | 2266.1 | scaffold split [40] |

## D.3 Baseline

We next briefly introduce the baseline explainers used in our experiments, including the gradients-based, the perturbation-based, and the surrogate GNN explanation methods.

- **GradCAM** [21] is a gradients-based explanation method. It conducts a weighted summation on the last layer representation to obtain the significance scores. The weights are provided by gradient back-propagation.

- **IG** [22] is a gradients-based explanation method. It calculates the significance scores by conducting path integral of the input feature gradients.

- **GNNExplainer** [15] is a perturbation-based explanation method. It perturbs both the topological structure and the node features to calculate the significance scores and search for explanation subgraphs $G_S$ that maximize the mutual information between $G_S$ and full graph prediction $\hat{y}$.

- **PGExplainer** [13] is a perturbation-based explanation method. It perturbs the topology and employs a parametric neural network to calculate the significance scores, with the same objective function in GNNExplainer.

- **ReFine** [36] is a perturbation-based explanation method. It applies the pre-training and fine-tuning paradigm to the GNN explanation exploration problem, which promises to offer an all-round inspection of the GNN decision-making process from multi-granularity.

- **PGM-Explainer** [37] is a surrogate explanation method. It trains an interpretable Bayesian network to fit the predicted label of the GNN model.

### D.4   Base GNN model

The architectures and downstream task performances of the base GNN models which function as the third-party predictors in our experiments are reported in Table 5.

Table 5: The architectures and downstream task performances of the base GNN models.

| Dataset | Backbone | Layers | Test Acc |
|---------|----------|--------|----------|
| BA-3Motifs | LEconv [41] | 2 | 100.00 |
| Mutag | GINConv [42] | 2 | 83.63 |
| Ogbg-molhiv | GCNConv [43] | 5 | 98.13 |
| Ogbg-ppa | GCNConv | 5 | 60.46 |

## E   Ablation Study

In the ablation study, we create three variants of the full V-InfoR. **No-VAE** removes the robust graph information extractor, which explores explanations based on the latent representations from the GNN model to be explained. **No-GIB** replaces the GIB-based optimization objective function with the traditional sparsity-driven mutual information objective function. **No-VAE-GIB** drops both two proposed modules. We report the performance of full V-InfoR and three variants in Figure 6.

It shows that the performance drops significantly when removing either the robust graph representation extractor or the adaptive graph information constraint. The performance of No-VAE-GIB is the lowest among the variants. As shown in Figure 6, the performance of No-GIB degrades more significantly than No-VAE in all the cases. This implies that the severe corruptions can more significantly affect the GNN explainers than the minor corruptions, which is consistent with the preliminary result in Section 2.1. On the one hand, the improvements from No-VAE-GIB to No-GIB and No-VAE indicate that both of the two modules are helpful to explain the corrupted graphs. On the other hand, the inferiority of No-GIB and No-VAE to the full V-InfoR further reveals that merely adopting one of the two modules is unable to deal with the graphs with two types of structural corruptions.

## F   Case Study

We give a visualized case study of explanations provided by V-InfoR and five baselines for the BA-3Motifs. Figure 7(a) shows the three types of motifs: cycle, grid, and house. Figure 7(b) shows the explanations on a graph with minor corruptions, whose prediction (cycle class) and ground-truth explanation (cycle) are both unchanged. V-InfoR allocates high significance scores (marked by red lines) to the cycle motif (highlighted by circle) correctly and eliminates the minor corruption successfully. While the other baseline explainers pay little attention to the vital cycle motif and fail to identify it as the explanation. Figure 7(c) shows the explanations of the severely corrupted graph. The raw prediction and raw ground-truth explanation are both associated with the grid motifs (marked by *"before corrupted"*), which have been broken by the severe corruption. Since the severe corruption has changed the prediction to house class, the explanation should be the house motif (marked by *"after corrupted"*). However, the five baseline explainers still allocate high significance scores to the component of the grid motif. Only V-InfoR succeeds in identifying the house motif (highlighted by circle) as the explanation.

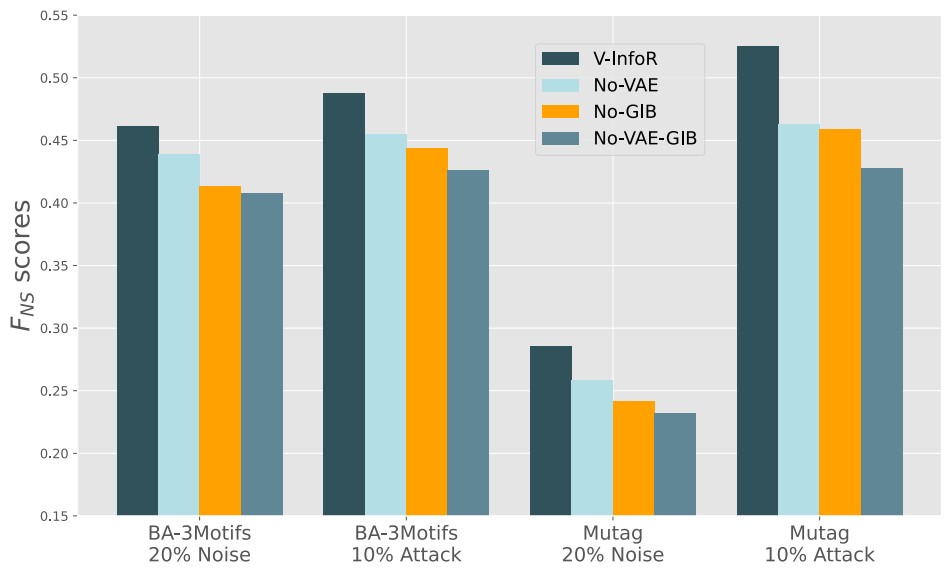

Figure 6: The ablation study of V-InfoR and three variants.

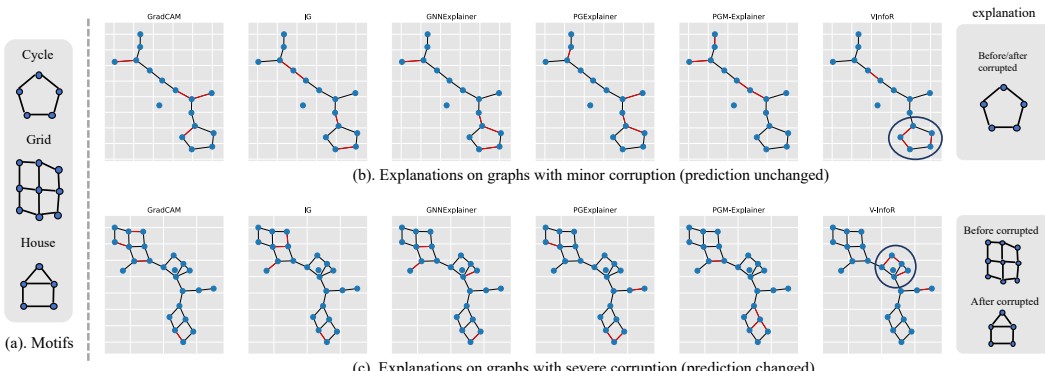

Figure 7: A case study of explanations provided by V-infoR and five baseline explainers.

# G  Robustness Evaluation

To directly demonstrate the robustness of our proposed V-InfoR, we introduce a robustness evaluation metric [44, 45], which is defined as the performance drop after a certain ratio of noise/adversarial attack is added into the graphs. A smaller performance drop means a more robust model and a negative value indicates that the model can even perform better under noise/adversarial attack.

First, we report the performance drop of V-InfoR and all the baselines in Table 6, with GRABNEL attack ratio set as {2.5%, 5.0%, 7.5%, 10%}. The results show that our proposed method can even improve the explanation performance on Mutag dataset after adversarial attack, while the performance of all the baselines degrade, which can demonstrate the robustness of our model to GRABNEL attack.

As shown in Table 7, we further report the performance drop of V-InfoR and all the baselines with the random noise ratio set as {0.05, 0.10, 0.15, 0.20, 0.25, 0.30}. For the Ogbg-molhiv and Ogbg-ppa, V-InfoR is more robust to the random noise than all the baselines. The result on Mutag indicates that V-InfoR stands among the top-tier robust methods, since the irregularity in chemical molecules is more obvious and significant. This phenomenon implies that our method relies more on capturing the irregularity stemming from graph corruption in order to maintain its robustness. When adding adversarial attack to Mutag graphs, the robustness of our method is more significant, which

Table 6: The robustness evaluation with GRABNEL attack.

| Dataset | Ratio | GradCAM | IG | GNNExplainer | PGExplainer | PGM-Explainer | ReFine | V-InfoR | Rank |
|---|---|---|---|---|---|---|---|---|---|
| Mutag | 2.5% | 0.0159 | 0.0190 | 0.0068 | 0.0056 | 0.0374 | 0.0092 | **-0.0054** | 1 |
| | 5.0% | 0.0038 | 0.0194 | 0.0287 | -0.0047 | 0.0245 | 0.0093 | **-0.0186** | 1 |
| | 7.5% | 0.0052 | 0.0286 | 0.0690 | -0.0032 | 0.0496 | 0.0274 | **-0.0172** | 1 |
| | 10.0% | 0.0027 | 0.0420 | 0.0593 | 0.0035 | 0.0637 | 0.0200 | **-0.0147** | 1 |
| BA-3Motifs | 2.5% | 0.0459 | 0.0141 | 0.0085 | 0.0157 | 0.0521 | 0.0504 | **-0.0047** | 1 |
| | 5.0% | 0.0668 | 0.0400 | 0.0554 | 0.0509 | 0.0557 | 0.0962 | **-0.0010** | 1 |
| | 7.5% | 0.2526 | 0.1926 | 0.0505 | 0.0700 | 0.1811 | 0.0996 | **0.0241** | 1 |
| | 10.0% | 0.3947 | 0.2608 | 0.1352 | **0.0859** | 0.2174 | 0.2521 | 0.1277 | 2 |

Table 7: The robustness evaluation with random noise.

| Dataset | Ratio | GradCAM | IG | GNNExplainer | PGExplainer | PGM-Explainer | ReFine | V-InfoR | Rank |
|---|---|---|---|---|---|---|---|---|---|
| Ogbg-molhiv | 0.05 | 0.0154 | 0.0741 | 0.0297 | 0.0247 | 0.0963 | 0.0318 | **0.0054** | 1 |
| | 0.10 | 0.0766 | 0.1469 | 0.0495 | 0.0577 | 0.0591 | 0.0554 | **0.0073** | 1 |
| | 0.15 | 0.0997 | 0.1914 | 0.0658 | 0.0569 | 0.0940 | 0.0605 | **0.0156** | 1 |
| | 0.20 | 0.1674 | 0.2132 | 0.1063 | 0.0518 | 0.1298 | 0.1325 | **0.0496** | 1 |
| | 0.25 | 0.2149 | 0.1606 | 0.1237 | 0.1237 | 0.1556 | 0.1692 | **0.0630** | 1 |
| | 0.30 | 0.2314 | 0.1775 | 0.1469 | 0.1593 | 0.1397 | 0.1890 | **0.0858** | 1 |
| Ogbg-ppa | 0.05 | 0.0045 | 0.0153 | 0.0063 | 0.0228 | 0.0537 | 0.0075 | **0.0013** | 1 |
| | 0.10 | 0.0506 | 0.0381 | 0.0402 | 0.1014 | 0.0789 | 0.0404 | **0.0172** | 1 |
| | 0.15 | 0.0797 | 0.0403 | 0.0535 | 0.1421 | 0.1331 | 0.0648 | **0.0237** | 1 |
| | 0.20 | 0.1010 | 0.0639 | 0.0844 | 0.2069 | 0.1049 | 0.0859 | **0.0328** | 1 |
| | 0.25 | 0.1357 | 0.1053 | 0.1093 | 0.2224 | 0.1591 | 0.1556 | **0.0254** | 1 |
| | 0.30 | 0.1797 | 0.0958 | 0.1267 | 0.2391 | 0.1656 | 0.1577 | **0.0491** | 1 |
| Mutag | 0.05 | 0.0866 | 0.0849 | 0.1037 | 0.0711 | 0.1131 | 0.0781 | **0.0647** | 1 |
| | 0.10 | 0.1669 | 0.1738 | 0.1532 | 0.1565 | 0.1986 | 0.1681 | **0.1189** | 1 |
| | 0.15 | 0.2656 | 0.2430 | 0.2348 | 0.2099 | 0.2647 | 0.2431 | **0.2097** | 1 |
| | 0.20 | 0.3218 | 0.3081 | 0.3023 | 0.2800 | 0.2794 | 0.2866 | **0.2403** | 1 |
| | 0.25 | 0.3417 | 0.3435 | 0.3343 | **0.3103** | 0.3359 | 0.3242 | 0.3118 | 2 |
| | 0.30 | 0.3655 | 0.3775 | 0.3819 | **0.3428** | 0.3690 | 0.3670 | 0.3452 | 2 |
| BA-3Motifs | 0.05 | -0.0074 | 0.0074 | 0.0020 | **-0.0137** | 0.0150 | 0.0231 | 0.0061 | 4 |
| | 0.10 | **-0.0041** | 0.0239 | 0.0019 | 0.0400 | 0.0353 | 0.0238 | 0.0163 | 3 |
| | 0.15 | 0.0825 | 0.0801 | **0.0493** | 0.0672 | 0.1043 | 0.0884 | 0.0878 | 5 |
| | 0.20 | **0.1125** | 0.1772 | 0.1358 | 0.1402 | 0.1874 | 0.1923 | 0.1527 | 4 |
| | 0.25 | **0.1278** | 0.2112 | 0.1574 | 0.1615 | 0.2100 | 0.2071 | 0.1836 | 4 |
| | 0.30 | **0.1407** | 0.2169 | 0.1617 | 0.1879 | 0.1952 | 0.2285 | 0.1951 | 4 |

verifies our deduction again. For BA-3Motifs, the result shows that our proposed model is within the middle range of the robustness evaluation on BA-3Motifs graphs with random noise. This can be attributed to two main possible reasons. Firstly, the introduction of random noise results in a lesser degree of irregularity compared to adversarial attack and thus the ability of our method to capture irregularities in corrupted graphs does not bring robustness improvement significantly. In contrast, our method showcases greater robustness than the baseline models when exposed to adversarial attacks on BA-3Motifs graphs. This is due to the fact that such attacks will destroy the influential motifs and consequently amplify the level of irregularity. Secondly, the synthetic BA-3Motifs graph itself adheres to a more regular structure than real-world graphs, which adopts a Barabasi-Albert graph as the base and attaches the base with one of three motifs. Consequently, the potential for our method to achieve robustness improvement is further constrained by the synthetic structural topology.

# H Limitation

**Node Feature Corruptions.** Both the robust graph representation extractor and the adaptive explanation generator are proposed to mitigate and address the graph structural corruptions. Thus, V-InfoR cannot be directly applied to explain the graphs with node feature corruptions. In the future work, we will extend the V-InfoR model to fulfill the robustness on node feature level.

**Multi-Task Explanation.** The explanation performance of V-InfoR explainer on some other graph tasks, such as node classification and link prediction tasks, has not been evaluated. We argue that V-InfoR can be directly used to explain the GNN predictor on node classification task, with little modification. However, the V-InfoR explainer for link prediction is non-trivial, since the GNN explanation methods on link prediction is scarce until now.

