# OpenReview forum: "V-InFoR: A Robust Graph Neural Networks Explainer for Structurally Corrupted Graphs"
_NeurIPS.cc/2023/Conference — NeurIPS 2023 poster_

### Official Review · Reviewer_SeJw · 2023-07-03

**Soundness:** 3 good
**Presentation:** 3 good
**Contribution:** 4 excellent
**Rating:** 8
**Confidence:** 4

**Summary:**

This paper proposed a robust GNN explaination method for structurally corrupted graphs. The novelty of the paper is that it considers the noise and corruptions in the graph, and verified that the corruptions can significantly affect the performance of existing GNN explainers. To address this issue, the authors proposed to use variational inference to sample a more robust graph representation and reduce the negative effect of the noise. The authors also formulate the problem as a graph information bottleneck optimization problem. Although straightforward, GIB method can effectively generate the explaination subgraph without any structural constrains, and thus is more general. The evaluations are extensively studied over one synthetic and three real graph datasets. By comparison with current SOTA models, the results show the significant performance improvement of the model in both minor and server corruptions.
Overall, the paper is very well written and has clear technique contributions. The evaluation is also convincing.


**Strengths:**

(1) The paper studied the GNN explanation problem when the graphs are noisy or corrupted, which is novel and practically important. In real application scenarios, the graphs may contain noise and not reliable, thus it is important to design a more robust explaination model for GNNs.
(2) The paper considers both mirror and severe corruptions on graph structures. To address the mirror corruptions, a variational inference based robust graph information extractor is proposed. An adaptive graph information constrained is also introduced into the GIB module. The proposed GIB optimization methods can capture both the regularity and irregularity of the corrupted graphs.
(3) The paper is well written and easy to read. A theoretical analysis is also provided to derive the variational bound of the objective function.
(4) The experiment is extensively conducted over both real and synthetic datasets. By comparison with current SOTA methods, the performance improvement of V-InfoR is significant.


**Weaknesses:**

(1) It seems that the robust graph representation extractor can effective filter minor noise. However, it the input graphs do not have minor noise, is this step still required? I wonder if the graph is reliable and do not contain much noise, whether the proposed method still works.
(2) The paper only considers the structural corruptions on the graphs. The graphs may also contain noise in the attributes. Can the methods be generalized to graphs with attributes corruptions? It would be better that the authors can further discuss how to deal with the attributes noise issue.


**Questions:**

(1) Does the model still work well for the graphs without much structural noise? A comparison between the proposed model and existing models on clean graphs would make the result more convincing.
(2) Although the authors focus on the structurally corrupted graphs, I still want to see whether the proposed method (like variational node representation) can be easily generalized to the attribute noise and corruptions?


**Limitations:**

Limitations are well discussed in the appendix, and I have no suggestions for improvement.

---

> ### Author Rebuttal · Authors · 2023-08-10
>
> >Q1: It seems that the robust graph representation extractor can effectively filter minor noise. However, if the input graphs do not have minor noise, is this step still required? I wonder if the graph is reliable and do not contain much noise, whether the proposed method still works. Does the model still work well for the graphs without much structural noise? A comparison between the proposed model and existing models on clean graphs would make the result more convincing.
>
> A1: Thanks for your comments. Actually, in Figure 3, the performance with zero noise-ratio corresponds to that on clean graphs. Here, we summarize the result in the following table.
> |Model|BA-3Motifs|Mutag|
> |:-|:-|:-|
> |GradCAM|0.5440|0.4889|
> |IG|0.5982|0.5077|
> |GNNExplainer|0.5363|0.4982|
> |PGExplainer|0.5150|0.4939|
> |PGM-Explainer|0.5287|0.5028|
> |ReFine|0.5912|0.5037|
> |Ours|**0.6157**(2.93%↑)|**0.5281**(4.02%↑)|
>
> The proposed method still achieves the best or top-tier performance when there are no corruptions. This reveals that even without any corruption, the robust representation extractor still extracts vital common graph information and the explanation generator adaptively identifies the explanations with regular structural properties.
> >Q2: The paper only considers the structural corruptions on the graphs. The graphs may also contain noise in the attributes. Can the methods be generalized to graphs with attributes corruptions? It would be better that the authors can further discuss how to deal with the attributes noise issue.
> Although the authors focus on the structurally corrupted graphs, I still want to see whether the proposed method (like variational node representation) can be easily generalized to the attribute noise and corruptions?
>
> A2: Thanks for your suggestion. The proposed method is able to generalize to the graphs with node attributes corruption and we may need to introduce some modules that is specially designed for the representation learning on node attributes. For example, we could filter the attribute noise and repair the node attribute based on the homophily assumption on node attributes. We leave the robust GNN explainer on node attributes corruption to the future work.

---

> > ### Comment · Reviewer_SeJw · 2023-08-14
> >
> > The authors have addressed my minor concerns in the rebuttal, and I will keep my socre as clear accept.

---

> > > ### Author Response · Authors · 2023-08-14
> > >
> > > Thank you again for your valuable comments and suggestions.

---

### Official Review · Reviewer_kswM · 2023-07-06

**Soundness:** 4 excellent
**Presentation:** 3 good
**Contribution:** 3 good
**Rating:** 7
**Confidence:** 4

**Summary:**

The study focuses on the significance of explaining Graph Neural Networks (GNN) models for the development of reliable and trustworthy machine learning systems. Unlike previous research that assumes clean input graphs, this paper introduces a novel approach to constructing a more robust GNN explainer capable of handling graphs with structural corruptions such as noisy and adversarial edges. This is a crucial consideration as real-world graphs are often imperfect. The authors specifically address two scenarios: minor corruptions that do not alter the GNN's prediction results, and severe corruptions that lead to changes in prediction outcomes. To tackle these scenarios, the proposed model, V-InfoR, consists of two modules. The robust graph representation extractor module aims to handle the first scenario, while the graph information bottleneck module focuses on addressing the second scenario. V-InfoR incorporating the two modules can explain well on both the corrupted graphs and the raw graphs. The effectiveness of the proposal is validated through evaluations conducted on four datasets.


**Strengths:**

S1: This paper explores the concept of graph structure corruptions at two different levels and highlights how both levels can negatively impact the quality of GNN explanations. This novel and intriguing study addresses a new and practically significant problem, considering that real-world graphs are susceptible to corruption and attacks.
S2: The use of variational inference to mitigate the effects of minor corruptions is a logical and robust approach. Additionally, the innovative GIB-based optimization framework effectively tackles the challenge of irregular severe corruptions. The utilization of the graph information bottleneck within the adaptive explanation generator appears to offer a more comprehensive and resilient model compared to existing methods. The integration of these two components allows for simultaneous handling of both types of structure corruptions. Overall, I think the proposed model is well-motivated and demonstrates technical solidity.
S3: The organization and writing of this paper are easy to follow. The motivation behind the research is also clearly conveyed by providing a preliminary experiment.
S4: The performance improvement of the proposed model, namely V-InfoR, is holistically verified on four datasets, including both the synthetic and real-world graphs.


**Weaknesses:**

W1: It is a little confusing that the primary results in experiment are presented and discussed in relation to random noise and adversarial attacks (referred to as adversarial edges in the main text). However, the authors establish definitions for both minor corruptions and severe corruptions. The authors may want to provide a clearer illustration of the relation and distinction between the two categories of corruptions (i.e., in experiment and in definition).
W2: I find the result presented in Figure 7 somewhat unclear, particularly regarding Figure 7c, where the performance of V-InfoR appears to improve after the graph is corrupted. This raises the question of why V-InfoR demonstrates better performance under such circumstances.


**Questions:**

Some questions are presented in the ‘Weaknesses’. Moreover, do the two terms in the total loss (Eq. (17)), i.e., LVAE and LGIB, need to be combined with a weighted hyper-parameter?

**Limitations:**

The authors have listed and discussed some potential limitations in the supplementary, including the node feature corruptions and the robust explanation on tasks besides graph classification. I wonder about the explanation efficiency of the proposed model.

---

> ### Author Rebuttal · Authors · 2023-08-10
>
> >Q1: It is a little confusing that the primary results in experiment are presented and discussed in relation to random noise and adversarial attacks (referred to as adversarial edges in the main text). However, the authors establish definitions for both minor corruptions and severe corruptions. The authors may want to provide a clearer illustration of the relation and distinction between the two categories of corruptions (i.e., in experiment and in definition).
>
> A1: Thanks for your suggestion. We will add more explanation about the relation and distinction between the two categories of corruptions in final version. In the preliminary experiment, we want to investigate the effect of minor and severe corruptions, respectively. Hence, we manually select the graph whose prediction is unchanged to simulate the minor corruption, as well as the simulation of severe corruption. However, in real-world, it is intractable to separate the minor and the severe corruptions, and both may exist simultaneously. Therefore, we use random noise and adversarial attack as structural corruptions which represent the corruption naturally exists in real-world and the man-made malicious corruption, respectively.
>
> >Q2: I find the result presented in Figure 7 somewhat unclear, particularly regarding Figure 7c, where the performance of V-InfoR appears to improve after the graph is corrupted. This raises the question of why V-InfoR demonstrates better performance under such circumstances.
>
> A2: Thanks for your comments. When explaining severely corrupted graphs, the explainer must identify the subgraph which conforms to the after-corrupted-prediction. In Figure 7c, the severe corruption has changed the prediction to house class. Therefore, the explanation should be the "house" motif marked by "after corrupted". Only V-InfoR succeeds in identifying the "house" motif explanation.
>
> >Q3: Do the two terms in the total loss (Eq. (17)), i.e., LVAE and LGIB, need to be combined with a weighted hyper-parameter?
>
> A3: Thanks for your comment. According to our experimental results, L-VAE and L-GIB are on the same order of magnitude. In practical use, it is also easy to introduce a weighted hyper-parameter, if the training process is unstable and hard to converge.

---

> > ### Comment · Reviewer_kswM · 2023-08-15
> > **reply**
> >
> > Thanks for the responses of the authors to my previous questions. My concerns are mostly addressed by their further explanation in the rebuttal, especially the clarification on two types of structural corruptions and further explanations on the result in Fig. 7. Generally I think this is a good paper with clear technique contributions. I like the idea of combining variational graph representation learning and the adaptive graph information bottleneck to address the GNN explanation problem with two types of noise. It is particularly interesting that the authors also investigate the issue that when the corruption can change the prediction result of the GNN, which is not considered by existing works.

---

> > > ### Author Response · Authors · 2023-08-15
> > >
> > > Thank you again for your valuable comments and detailed suggestions.

---

### Official Review · Reviewer_cjXo · 2023-07-06

**Soundness:** 3 good
**Presentation:** 3 good
**Contribution:** 2 fair
**Rating:** 6
**Confidence:** 4

**Summary:**

This work aims at structurally corrupted graphs for a more robust explanation. It proposes a method called V-InFoR, which includes two components for minor and severe corruptions respectively. Experiments on synthetic and real data with different corruption ratios verify the effectiveness of the proposal.

**Strengths:**

1. The motivation is clearly presented by a pre-experiment. And the overall structure of the paper is easy to follow.
2. The two components (VAE and GIB) of the proposed method are not trivial, especially the distribution representation may not be easy to train.
3. The experimental results are convincing, especially those under adversarial attacks.

**Weaknesses:**

1. The authors provide several simulated examples of corruption, which is good. However, they could present more practical examples of corruption in the real world, just at the data level, which could be better for readers to understand the need for robustness.
2. The motivation for the introduction of GIB beyond VAE should be further clarified (lines 180-183), as there is no precise dividing line for minor and severe corruption.

**Questions:**

1. Major
    1. In the ablation studies, GIB is more effective in both minor and major corruption cases - any further explanation or intuition behind this phenomenon is welcome.
    2. For the results in Figure 3, further ablation studies of VAE and GIB are recommended.
2. Minor
    1. In Figure 2, the VAE loss should also be noted.

If I have missed anything, please let me know.

**Limitations:**

Limitations are included in the appendix.

---

> ### Author Rebuttal · Authors · 2023-08-10
>
> >Q1: The authors provide several simulated examples of corruption, which is good. However, they could present more practical examples of corruption in the real world, just at the data level, which could be better for readers to understand the need for robustness.
>
> A1: Thanks for your suggestion. We will add a few practical examples of corruptions in the final version.
> 1). On one hand, the raw data usually contain some noise naturally in real applications, and thus a robust model that considers the effect of such random noise is needed. In this case, our method can get improvement even when no extra noise is manually added. As shown in Fig 3, our method improves the model performance compared with baselines on the raw graphs without any noise added (noise ratio is 0%).
> 2). On the other hand, the adversaries are common and the fake data can be easily injected into the raw data [1, 2], such as recommender systems, knowledge graphs, and social networks. For example, the fraudsters frequently manipulate online reviews and product websites and the spammers add fake connections to social networks.
> >Q2: The motivation for the introduction of GIB beyond VAE should be further clarified (lines 180-183), as there is no precise dividing line for minor and severe corruption.
>
> A2: Thanks for your comment. The fact is that the minor and severe corruptions are hard to separate and tend to exist simultaneously. Therefore, we need to combine VAE and GIB together to comprehensively mitigate both the minor and severe corruptions. As the ablation study shows, the explanation performance degrades when removing the GIB module. Moreover, we notice that the performance degradation on Mutag dataset increases from 1.13\% to 5.42\% as the noise ratio growing from 5\% to 15\%, which demonstrates the necessity of GIB, especially when explaining severely corrupted graphs.
> >Q3: In the ablation studies, GIB is more effective in both minor and major corruption cases - any further explanation or intuition behind this phenomenon is welcome.
>
> A3: Thanks for your suggestion. Actually, the “Noise” and “Attack” situation in the ablation studies may not totally correspond to the “minor” and “major” corruption, respectively. In real application scenarios, it is intractable to separate the minor and the severe corruptions, and both may exist simultaneously. Therefore, the effectiveness of GIB may imply that the severe corruptions can more significantly affect the GNN explainers than the minor corruptions. Furthermore, the information constraint term MI$(G,G_S)$ in GIB also functions as the sparsity regularization. This additional function may improve the explainer performance too.
> >Q4: For the results in Figure 3, further ablation studies of VAE and GIB are recommended.
>
> A4: Thanks for your suggestion. We will add more ablation studies in Appendix G of the final version. Specifically, we report the performance of the three variants (i.e., No-VAE, No-GIB, and No-VAE-GIB) under different noise ratios. The Fns-score on BA-3Motifs is listed in the following table.
> |Noise Ratio|V-InfoR|No-VAE|No-GIB|No-VAE-GIB|
> |:-|:-|:-|:-|:-|
> |0|0.6167|0.6003|0.5994|0.5956|
> |0.05|0.6076|0.5891|0.5703|0.5599|
> |0.1|0.5974|0.5530|0.5229|0.5164|
> |0.15|0.5239|0.5115|0.4827|0.4625|
> |0.2|0.4610|0.4387|0.4136|0.4080|
> |0.25|0.4310|0.4139|0.3894|0.3826|
> |0.3|0.4154|0.3921|0.3717|0.3672|
>
> The Fns-score on Mutag is listed in the following table.
> |Noise Ratio|V-InfoR|No-VAE|No-GIB|No-VAE-GIB|
> |:-|:-|:-|:-|:-|
> |0|0.5280|0.5107|0.5112|0.5064|
> |0.05|0.4608|0.4529|0.4495|0.4423|
> |0.1|0.4066|0.3833|0.3746|0.3711|
> |0.15|0.3158|0.2848|0.2616|0.2528|
> |0.2|0.2852|0.2586|0.2415|0.2341|
> |0.25|0.2173|0.1969|0.1774|0.1632|
> |0.3|0.1803|0.1627|0.1439|0.1408|
>
> The result shows that the performance of No-GIB degrades more significantly than No-VAE in all the cases. This implies that the severe corruption affects the GNN explanation more significantly than the minor corruption. Moreover, the inferiority of No-GIB and No-VAE to the full V-InfoR further reveals that merely adopting one of the two modules is unable to deal with the graphs with two types of structural corruptions.
> >Q5: In Figure 2, the VAE loss should also be noted.
>
> A5: Thanks for your suggestion. In the final version, we will note the VAE loss in Figure 2.
>
> [1] Adversarial Attacks on Neural Networks for Graph Data. KDD 2018.
>
> [2] NetFense: Adversarial Defenses Against Privacy Attacks on Neural Networks for Graph Data. TKDE 2023.

---

> > ### Comment · Reviewer_cjXo · 2023-08-14
> > **Thanks for the rebuttal**
> >
> > We thank the authors for their detailed rebuttal, especially the attached ablation experiments. Also, the answer to Q3 is necessary to understand the improvement gap between two models that agree with the ablation results.
> > Overall, I am happy to raise my score to 6 and good luck.

---

> > > ### Author Response · Authors · 2023-08-14
> > >
> > > We are grateful for your detailed comments and insightful suggestions.
> > >
> > > Thank you again for your willingness to raise the score.

---

### Official Review · Reviewer_2tSN · 2023-07-07

**Soundness:** 3 good
**Presentation:** 3 good
**Contribution:** 3 good
**Rating:** 5
**Confidence:** 5

**Summary:**

This work introduces a robust GNN explainer for structurally-corrupted graphs. The key idea is to exploit the statistics of the graph's latent distributions insensitive to structural perturbations for reconstructing a new graph that will be further leveraged for subgraph (GNN explainer) identification. Through extensive experiments, the proposed approach is shown to be more robust than existing GNN explainers under strong graph structural perturbations.

**Strengths:**

1) The idea of using GNNs to extract the graph's latent distributions for graph reconstruction is interesting and effective.
2) The proposed adaptive explanation generator is designed based on a theoretically-sound variational upper bound, which also shows its effectiveness in extensive experimental results.

**Weaknesses:**

1) There should be more discussions of existing works that aim to reconstruct graphs based on properties insensitive to structural perturbations. For example, the graph spectral properties (e.g. the first few Laplacian eigenvalues/eigenvectors) are not sensitive to adversarial attacks and thus can be leveraged for graph reconstruction (see a recent work "GARNET: Reduced-Rank Topology Learning for Robust and Scalable Graph Neural Networks" published in LoG'22).
2) There should be a section discussing the algorithm runtime/space complexity of the proposed framework. It is not clear if this method can be efficiently applied to large graphs.

**Questions:**

The statistical metrics mean and deviation is considered for extracting the latent distribution. Will other higher-order statistical metrics help in this step?


**Limitations:**

There should be a section discussing the runtime/space complexity of the proposed method. Additional runtime results should also be reported for graphs with different sizes.

---

> ### Author Rebuttal · Authors · 2023-08-10
>
> >Q1: There should be more discussions of existing works that aim to reconstruct graphs based on properties insensitive to structural perturbations. For example, the graph spectral properties (e.g., the first few Laplacian eigenvalues/eigenvectors) are not sensitive to adversarial attacks and thus can be leveraged for graph reconstruction (see a recent work "GARNET: Reduced-Rank Topology Learning for Robust and Scalable Graph Neural Networks" published in LoG'22).
>
> A1: Thanks for your suggestion. We will add the suggested reference and provide more discussions on the graph reconstruction methods. When explaining the structurally corrupted graphs, it is promising to defense the structural perturbations based on the graph spectral properties in GARNET [1]. As we state in line 46-56 of the paper, the structural corruption includes both the minor corruption and severe corruption, based on whether the prediction of downstream task is changed or not. For the minor corruption, introducing graph reconstruction methods as a noise filter may also be a reasonable solution, since the minor corruption does not affect the final prediction and the GNN explainer can identify the corresponding explanation. But, for the severe corruption, the final prediction has been changed. If the GNN explainer searches for the explanation based on the reconstructed graph, it cannot be guaranteed that the explanation is corresponding to the model prediction, since the reconstructed graph may have eliminated the severe corruption and recorrected the prediction.
> >Q2: There should be a section discussing the algorithm runtime/space complexity of the proposed framework. It is not clear if this method can be efficiently applied to large graphs. Additional runtime results should also be reported for graphs with different sizes.
>
> A2: Thanks for your comments. Our method can be efficiently applied to large graph. We will add a section to analyze the time/space complexity of the proposed model V-InfoR. The time complexity of V-InfoR to explain a new graph with $|V|$ nodes and $|E|$ edges is $O(|V|^2+|E|)$, which is similar to the previous works PGExplainer [2] ($O(|V|^2+|E|)$) and GNNExplainer [3] ($O(|V|^2+k*|E|)$). The space complexity is $O(|V|+|E|)$, since we need to restore the node representations. The average runtime on a single graph with a single NVIDIA GeForce 3060 GPU (6GB) is reported in the following table.
> |Dataset|Avg. #Nodes|Avg. #Edges|Avg. Runtime(ms)|
> |:-|:-|:-|:-|
> |BA-3Motifs|21.92|14.76|20.81|
> |Mutag|30.32|30.77|38.76|
> |Ogbg-molhiv|25.50|27.50|29.42|
> |Ogbg-ppa|243.40|2266.10|2584.60|
>
> >Q3: The statistical metrics mean and deviation is considered for extracting the latent distribution. Will other higher-order statistical metrics help in this step?
>
> A3: Thanks for your comment. When it comes to variational auto-encoder, we usually assume that the variational distribution belongs to Gaussian distribution. A Gaussian distribution can be determined by the mean $\mu$ and variation $\sigma^2$ and thus generally the mean and variation (deviation) are enough to deduce the latent distribution [4]. However, in some complex situation, Gaussian distribution may not be a reasonable assumption and then introduce some high-order statistics could achieve better performance. We will investigate the effect of the higher-order statistics in the future work.
>
> [1] Chenhui Deng, Xiuyu Li, Zhuo Feng, Zhiru Zhang. GARNET: Reduced-Rank Topology Learning for Robust and Scalable Graph Neural Networks. LoG 2022.
>
> [2] Parameterized Explainer for Graph Neural Network. NeurIPS 2020.
>
> [3] GNNExplainer: Generating Explanations for Graph Neural Networks. NeurIPS 2019.
>
> [4] Thomas N. Kipf, Max Welling. Variational Graph Auto-Encoders. NIPS 2016.

---

> > ### Comment · Reviewer_2tSN · 2023-08-14
> >
> > Thank you for your response. However, I will keep my original score.

---

> > > ### Author Response · Authors · 2023-08-14
> > >
> > > Thanks for your valuable comments and suggestions again.

---

### Official Review · Reviewer_Q8Xb · 2023-07-18

**Soundness:** 2 fair
**Presentation:** 1 poor
**Contribution:** 2 fair
**Rating:** 6
**Confidence:** 4

**Summary:**

This paper proposes a method of explaining GNNs when the input graph is noisy. Existing methods tend to misexplain when noisy graphs are given. The proposed method uses GraphVAE to remove minor noise from the input graph, and a formulation with information bottlenecks allows detecting the presence of severe noise. Experiments confirm the effectiveness of the proposed method using one artificial and three real datasets.

**Strengths:**

- This paper is the first to propose a GNN explanation method in noisy settings. This problem setting seems important in practice.
- The proposed method consistently outperforms existing methods in the experiments.

**Weaknesses:**

- I was not convinced that the GIB loss is particularly effective in noisy situations, although it is indeed a plausible explanation. More detailed discussions are required.
- Figure 3 shows that the proposed method outperforms the existing methods by the same amount in the absence of noise and in the presence of noise, and when noise is added, the accuracy drops at the same rate as the existing methods. The reason the proposed method outperforms the existing method may not be because it is robust to noise, but simply because it is more accurate. High accuracy is great, but given the subject of this work, which focuses on noise, it is not aligned with the main claim.
- The proposed method is too complex for practical use. I would not want to use this as a practitioner.
- The experiment only uses GRABNEL as severe noise. Although the authors state that GRABNEL is the attack method of SoTA, there are many aspects of the attack, and defense against only GRABNE is not sufficient. It would be better to use more attack methods, including basic ones such as nettack and metattack.
- The method section is messy. It would be helpful to summarize a list of parameters to be optimized at the end.

**Questions:**

- Could you provide an intuitive explanation on why the GIB loss is robust with respect to severe noise?
- Based on the trends in Figure 3, the proposed method may not be robust to noise, but simply accurate. Could you show that the proposed method is indeed robust to noise?

**Limitations:**

Potential negative societal impacts do not apply.

---

> ### Author Rebuttal · Authors · 2023-08-10
>
> >Q1: Detailed discussion and intuitive explanation on the robustness of GIB loss.
>
> A1: Thanks for your comments.
>
> 1). The severe corruption should be identified as a part of the explanation, since it is the non-ignorable reason that changes the model prediction. However, the irregularity issue of severe corruption makes it difficult for previous methods to identify the severe corruption as part of the explanation. Therefore, we propose the GIB loss to address the irregularity issue and thus make the proposed method robust to severely corrupted graphs.
>
> 2). For example, an important motif may be corrupted by adversarial attack and becomes unconnected (removing edges) or overly large (adding edges). In this case, the regularity constraints on the explanation graph size or connectivity that adopted by previous methods are not plausible anymore. Different from them, the GIB loss does not need any regularity constraint and directly restricts the explanation by minimizing the mutual information between the explanation and the raw graph, which is able to identify the influential subgraph under severe corruption.
> >Q2: Could you show that the proposed method is indeed robust to noise?
>
> A2: Thanks for your comments. In Figure 3, as the noise ratio growing, the explanation problem also becomes more difficult and thus the performance of all the explainers (including ours) drops. Our method is able to achieve SOTA performance compared with all the baselines consistently when different amounts of noise are added into the raw graphs. Although the overall performance trend of our method is similar to baselines, its performance improvements over the two strong baselines (i.e., ReFine and IG) differ with different noise ratios.
>
> Specifically, compared with ReFine, the improvement of our method is 3.00\% when only 10\% noise is added but the improvement increases to 6.21\% when 20\% noise is added. It implies that compared with ReFine, the performance improvement of our method is more significantly when more noise is added into the graphs (from 10\% to 20\%). Compared with IG, the improvement increases from 0.25\% to 4.31\% as the noise ratio growing from 15\% to 25\%.  To sum up, when the explanation problem grows hard, our method can maintain or even further boost the outperformance, which is able to demonstrate its robustness to graph structural corruptions.
> >Q3: Complexity of the proposed method for practical use.
>
> A3: Thanks for your comment. Actually, our method is easy for practical use, with the open-source code at https://anonymous.4open.science/r/V-InfoR-EF88. Compared with previous GNN explanation methods [1, 2], the proposed method has three main hyperparameters $\pi$, $\beta$, and $\tau$ need to initialize, but these parameters are not hard to tune. We can use bash scripts to modify the three hyperparameters for convenience. Specifically, $\pi$ in Formula 16 represents the prior Bernoulli probability, $\beta$ in Formula 8 controls the balance between the two terms of GIB loss, and $\tau$ in Formula 10 controls the approximation degree to Bernoulli distribution. According to our experimental results, users can focus on tuning $\pi$ on specific dataset. As the hyper-parameter analysis shows, our method is not sensitive to $\beta$ and the recommended value of $\tau$ is 0.3.
>
> The time complexity of V-InfoR to explain a new graph with $|V|$ nodes and $|E|$ edges is $O(|V|^2+|E|)$, which is similar to the previous works PGExplainer ($O(|V|^2+|E|)$) and GNNExplainer ($O(|V|^2+k*|E|)$). The space complexity is $O(|V|+|E|)$, since we need to restore the node representations.
>
> If the users want to use our model to explain a new GNN that trained on his own dataset, the following instruction may be helpful. First, initialize the hyperparameters. Second, train our explainer on his own training set and the GNN model functions as an oracle. Third, tune the hyperparameter based on the validation set. Finally, input the new instance to the trained explainer and the corresponding explanation is provided.
> >Q4: Evaluation on more attack methods.
>
> A4: Thanks for your suggestion. We have added the comparison experiment against nettack [3] and metattack [4] on Mutag dataset. The results are listed in the following table.
> |Nettack|5%|10%|
> |:-|:-|:-|
> |GradCAM|0.5035|0.4392|
> |IG|0.5362|0.5064|
> |GNNExplainer|0.5142|0.4315|
> |PGExplainer|0.4977|0.4916|
> |PGM-Explainer|0.5156|0.4831|
> |ReFine|0.5398|0.5120|
> |Ours|**0.5500**(1.89%↑)|**0.5263**(2.79%↑)|
>
> |Metattack|5%|10%|
> |:-|:-|:-|
> |GradCAM|0.4909|0.4459|
> |IG|0.5220|0.4705|
> |GNNExplainer|0.5018|0.4794|
> |PGExplainer|0.4813|0.4841|
> |PGM-Explainer|0.4864|0.4671|
> |ReFine|0.5057|0.4925|
> |Ours|**0.5487**(5.11%↑)|**0.5215**(5.89%↑)|
>
> The result shows that our method can also consistently outperform the baselines, for the adversarial attack algorithm nettack and metattack.
> >Q5: Summarize a list of parameters to be optimized at the end.
>
> A5: Thanks for your suggestion. We will improve the organization and writing of the method section in the final version if the paper is accepted. Here, we add a summarization of parameters to be optimized, as the table follows.
> |Notation|Description|1st Appearance|
> |:-|:-|:-|
> |$\beta$|Weighted parameter between the two terms of GIB loss|Eq.8|
> |$\tau$|Approximation degree to Bernoulli distribution|Eq.10|
> |$\pi$|Prior Bernoulli probability|Eq.16|
>
> **We also summarize the basic notations and descriptions in our paper, please see Appendix B for more details.**
>
> [1] Parameterized Explainer for Graph Neural Network. NeurIPS 2020.
>
> [2] GNNExplainer: Generating Explanations for Graph Neural Networks. NeurIPS 2019.
>
> [3] Adversarial Attacks on Neural Networks for Graph Data. KDD 2018.
>
> [4] Adversarial Attacks on Graph Neural Networks via Meta Learning. ICLR 2019.

---

> > ### Comment · Reviewer_Q8Xb · 2023-08-17
> >
> > Thank you for the reply.
> >
> > > Q2
> >
> > > Specifically, compared with ReFine, the improvement of our method is 3.00% when only 10% noise is added but the improvement increases to 6.21% when 20% noise is added. It implies that compared with ReFine, the performance improvement of our method is more significantly when more noise is added into the graphs (from 10% to 20%). Compared with IG, the improvement increases from 0.25% to 4.31% as the noise ratio growing from 15% to 25%.
> >
> > I'm concerned that these values were cherry-picked. Figure 3 (a) says the improvement from GramCAM decreases as the noise ratio increases. Figure 3 (b) says that the improvement from PGExplainer does not change much when the noise ratio becomse 0.3 from 0.0. The additional experiments (Q4) say PGExplainer is more robust to noise.
> >
> > This point is critical to support the main claim (i.e., noise robustness) of the proposed method.
> >
> > > Q3
> >
> > I still think the proposed method is too complex and cumbersome in practice. I (and many practitioners) would not want to train an additional VAE to interpret a model.
> >
> > I maintain the score due to the above concerns.

---

> > > ### Author Response · Authors · 2023-08-18
> > > **Further Rebuttal by Authors**
> > >
> > > >Q: I'm concerned that these values were cherry-picked. Figure 3 (a) says the improvement from GramCAM decreases as the noise ratio increases. Figure 3 (b) says that the improvement from PGExplainer does not change much when the noise ratio becomse 0.3 from 0.0. The additional experiments (Q4) say PGExplainer is more robust to noise.
> > > This point is critical to support the main claim (i.e., noise robustness) of the proposed method.
> > >
> > > A: Thanks for your further comment.
> > > We think the reviewer and us may have some gaps on the understanding of the robustness. We totally agree that if with the increase of the noise ratio, the performance of a model will not decrease significantly and finally become stable is definitely meaning a robust model. However, in many real application scenarios, the cases are complex and the robustness of a model is not always the case as described above. We think a model can also be considered to be robust if adding different noise ratios and different types of noise to the raw data, its performance can be consistently better than other models and its improvement can be kept even when the noise ratio is high (like 30\%).
> > >
> > > This is also a common scenario as shown in some previous works [1, 2]. For example, in work [1], the authors proposed a robust GCN model, and its performance curve in the result figures (Figures 2 and 3 in their paper) present similar decreasing trend to baseline performance curves but always above them, which is quite similar to the result figure (Figure 3) in our paper. Similar understanding on robustness also appears in CV area paper [2].
> > > >Q: I still think the proposed method is too complex and cumbersome in practice. I (and many practitioners) would not want to train an additional VAE to interpret a model.
> > >
> > > A: Thanks for your further comment. Our proposed model is trained end-to-end and thus the VAE module does not need an additional training process.
> > >
> > > [1] Robust Graph Convolutional Networks Against Adversarial
> > > Attacks. KDD 2019.
> > >
> > > [2] Certified Patch Robustness via Smoothed Vision Transformers. CVPR 2022.

---

> > > > ### Comment · Reviewer_Q8Xb · 2023-08-18
> > > >
> > > > > We think a model can also be considered to be robust if adding different noise ratios and different types of noise to the raw data, its performance can be consistently better than other models and its improvement can be kept even when the noise ratio is high (like 30%).
> > > >
> > > > I disagree with this point. Your claim would conclude ResNet is more noise robust than AlexNet, and transformers are more noise robust than RNNs, and Adam makes models more noise robust than SGD. I don't think so. They are irrelevant to noise robustness. ResNet just performs better than AlexNet in many scenarios. It does not mean ResNet is more noise robust.
> > > >
> > > > Similarly, the proposed method just performs better than the existing methods. It does not corroborate the proposed method is more noise robust.
> > > >
> > > > Performing well is a good thing, but it does not support the main claim (i.e., noise robustness) of the proposed method.

---

> > > > > ### Author Response · Authors · 2023-08-20
> > > > > **Further Rebuttal by Authors**
> > > > >
> > > > > Thanks again for your further comment. Your comment is greatly helpful to improve our paper. Following your comment, to demonstrate our proposed method is robust to noise/adversarial attack, we add a new robustness evaluation metric, which is defined as the performance drop after a certain ratio of noise/adversarial attack is added into the graphs, following many existing works [1] [2] [3]. A smaller performance drop means a more robust model and a negative value indicates that the model can even perform better under noise/adversarial attack.
> > > > >
> > > > > 1). We report the result with the new robustness metric on Ogbg-molhiv and Ogbg-ppa datasets with the random noise ratio set as 0.05, 0.1, 0.15, 0.2, 0.25 and 0.3. The results are shown in the following tables, where the highest score is marked with bold font and the second highest score is marked with underline.
> > > > > Ogbg-molhiv|GradCAM|IG|GNNExplainer|PGExplainer|PGM-Explainer|ReFine|V-InfoR (Ours)|Rank
> > > > > -|-|-|-|-|-|-|-|-
> > > > > 0.05|$\underline{0.0154}$|$0.0741$|$0.0297$|$0.0247$|$0.0963$|$0.0318$|$\mathbf{0.0054}$|1
> > > > > 0.1|$0.0766$|$0.1469$|$\underline{0.0495}$|$0.0577$|$0.0591$|$0.0554$|$\mathbf{0.0073}$|1
> > > > > 0.15|$0.0997$|$0.1914$|$0.0658$|$\underline{0.0569}$|$0.0940$|$0.0605$|$\mathbf{0.0156}$|1
> > > > > 0.2|$0.1674$|$0.2132$|$0.1063$|$\underline{0.0518}$|$0.1298$|$0.1325$|$\mathbf{0.0496}$|1
> > > > > 0.25|$0.2149$|$0.1606$|$\underline{0.1237}$|$\underline{0.1237}$|$0.1556$|$0.1692$|$\mathbf{0.0630}$|1
> > > > > 0.3|$0.2314$|$0.1775$|$0.1469$|$0.1593$|$\underline{0.1397}$|$0.1890$|$\mathbf{0.0858}$|1
> > > > >
> > > > > Ogbg-ppa|GradCAM|IG|GNNExplainer|PGExplainer|PGM-Explainer|ReFine|V-InfoR (Ours)|Rank
> > > > > -|-|-|-|-|-|-|-|-
> > > > > 0.05|$\underline{0.0045}$|$0.0153$|$0.0063$|$0.0228$|$0.0537$|$0.0075$|$\mathbf{0.0013}$|1
> > > > > 0.1|$0.0506$|$\underline{0.0381}$|$0.0402$|$0.1014$|$0.0789$|$0.0401$|$\mathbf{0.0172}$|1
> > > > > 0.15|$0.0797$|$\underline{0.0403}$|$0.0535$|$0.1421$|$0.1331$|$0.0648$|$\mathbf{0.0237}$|1
> > > > > 0.2|$0.1010$|$\underline{0.0639}$|$0.0844$|$0.2069$|$0.1049$|$0.0859$|$\mathbf{0.0328}$|1
> > > > > 0.25|$0.1357$|$\underline{0.1053}$|$0.1093$|$0.2224$|$0.1591$|$0.1556$|$\mathbf{0.0254}$|1
> > > > > 0.3|$0.1797$|$\underline{0.0958}$|$0.1267$|$0.2391$|$0.1656$|$0.1577$|$\mathbf{0.0491}$|1
> > > > >
> > > > > The results above can demonstrate that our proposed model is more robust to the random noise than all the baselines, on Ogbg-molhiv and Ogbg-ppa datasets.
> > > > >
> > > > > 2). We summarize the results in Table 2 of our paper and present the robustness evaluation on BA-3Motifs and Mutag datasets with the adversarial attack budget ranging from 2.5% to 10%. The results are shown in the following tables,where the highest score is marked with bold font and the second highest score is marked with underline.
> > > > > BA-3Motifs|GradCAM|IG|GNNExplainer|PGExplainer|PGM-Explainer|ReFine|V-InfoR (Ours)|Rank
> > > > > -|-|-|-|-|-|-|-|-
> > > > > 2.5%|$0.0459$|$0.0141$|$\underline{0.0085}$|$0.0157$|$0.0521$|$0.0504$|$\mathbf{-0.0047}$|1
> > > > > 5%|$0.0668$|$\underline{0.0400}$|$0.0554$|$0.0509$|$0.0557$|$0.0962$|$\mathbf{0.0010}$|1
> > > > > 7.5%|$0.2526$|$0.1926$|$\underline{0.0505}$|$0.0700$|$0.1811$|$0.0996$|$\mathbf{0.0241}$|1
> > > > > 10%|$0.3947$|$0.2608$|$0.1352$|$\mathbf{0.0859}$|$0.2174$|$0.2521$|$\underline{0.1277}$|2
> > > > >
> > > > > Mutag|GradCAM|IG|GNNExplainer|PGExplainer|PGM-Explainer|ReFine|V-InfoR (Ours)|Rank
> > > > > -|-|-|-|-|-|-|-|-
> > > > > 2.5%|$0.0159$|$0.0190$|$0.0068$|$\underline{0.0056}$|$0.0374$|$0.0092$|$\mathbf{-0.0054}$|1
> > > > > 5%|$0.0038$|$0.0194$|$0.0287$|$\underline{-0.0047}$|$0.0245$|$0.0093$|$\mathbf{-0.0186}$|1
> > > > > 7.5%|$0.0052$|$0.0286$|$0.0690$|$\underline{-0.0032}$|$0.0496$|$0.0274$|$\mathbf{-0.0172}$|1
> > > > > 10%|$\underline{0.0027}$|$0.0420$|$0.0593$|$0.0035$|$0.0637$|$0.0200$|$\mathbf{-0.0147}$|1
> > > > >
> > > > > The results show that our proposed method can even improve the explanation performance on Mutag dataset after adversarial attack, while the performance of all the baselines degrade, which can demonstrate the robustness of our model to adversarial attack.
> > > > >
> > > > > Thanks again for your insightful comments, which we believe are very important to improve our paper. We will add more experiments and discussions in the final version to support the robustness of our proposed method.
> > > > >
> > > > > [1] All You Need is Low (Rank): Defending Against Adversarial Attacks on
> > > > > Graphs. WSDM 2020.
> > > > >
> > > > > [2] Graph Structure Learning for Robust Graph Neural Networks. KDD 2020.
> > > > >
> > > > > [3] Graph Adversarial Training: Dynamically Regularizing Based on Graph Structure. TKDE 2021.

---

> > > > > > ### Comment · Reviewer_Q8Xb · 2023-08-21
> > > > > >
> > > > > > Thank you for the additional reports. This robustness metric is much better than what was used in the submission.
> > > > > >
> > > > > > My concern is that these results were cherry-picked.
> > > > > > Could you report the tables for BA-3Motifs and Mutag with the random noise ratio set as 0.05, 0.1, 0.15, 0.2, 0.25 and 0.3 with the same format and metric?

---

> > > > > > > ### Author Response · Authors · 2023-08-21
> > > > > > >
> > > > > > > Thank you for the further comments. Here, we report the robustness metric on BA-3Motifs and Mutag datasets with the random noise ratio set as 0.05, 0.1, 0.15, 0.2, 0.25 and 0.3.
> > > > > > >
> > > > > > > 1). The robustness metric on Mutag dataset is shown in the following table, where the highest score is marked with bold font and the second highest score is marked with underline.
> > > > > > > Mutag|GradCAM|IG|GNNExplainer|PGExplainer|PGM-Explainer|ReFine|V-InfoR (Ours)|Rank
> > > > > > > -|-|-|-|-|-|-|-|-
> > > > > > > $0.05$|$0.0866$|$0.0849$|$0.1037$|$\underline{0.0711}$|$0.1131$|$0.0781$|$\mathbf{0.0647}$|$1$
> > > > > > > $0.1$|$0.1669$|$0.1738$|$\underline{0.1532}$|$0.1565$|$0.1986$|$0.1681$|$\mathbf{0.1189}$|$1$
> > > > > > > $0.15$|$0.2656$|$0.2430$|$0.2348$|$\underline{0.2099}$|$0.2647$|$0.2431$|$\mathbf{0.2097}$|$1$
> > > > > > > $0.2$|$0.3218$|$0.3081$|$0.3023$|$0.2800$|$\underline{0.2794}$|$0.2866$|$\mathbf{0.2403}$|$1$
> > > > > > > $0.25$|$0.3417$|$0.3435$|$0.3343$|$\mathbf{0.3103}$|$0.3359$|$0.3242$|$\underline{0.3118}$|$2$
> > > > > > > $0.3$|$0.3655$|$0.3775$|$0.3819$|$\mathbf{0.3428}$|$0.3690$|$0.3670$|$\underline{0.3452}$|$2$
> > > > > > >
> > > > > > > The result indicates that V-InfoR stands among the top-tier robust methods, since the irregularity in chemical molecules is more obvious and significant. This phenomenon implies that our method relies more on capturing the irregularity stemming from graph corruption in order to maintain its robustness. When adding adversarial attack to Mutag graphs, the robustness of our method is more significant, which verifies our deduction again.
> > > > > > >
> > > > > > > 2). The robustness metric on BA-3Motifs dataset is shown in the following table.
> > > > > > > BA-3Motifs|GradCAM|IG|GNNExplainer|PGExplainer|PGM-Explainer|ReFine|V-InfoR (Ours)|Rank
> > > > > > > -|-|-|-|-|-|-|-|-
> > > > > > > $0.05$|$\underline{-0.0074}$|$0.0074$|$0.0020$|$\mathbf{-0.0137}$|$0.0150$|$0.0231$|$0.0061$|$4$
> > > > > > > $0.1$|$\mathbf{-0.0041}$|$0.0239$|$\underline{0.0019}$|$0.0400$|$0.0353$|$0.0238$|$0.0163$|$3$
> > > > > > > $0.15$|$0.0825$|$0.0801$|$\mathbf{0.0493}$|$\underline{0.0672}$|$0.1043$|$0.0884$|$0.0878$|$5$
> > > > > > > $0.2$|$\mathbf{0.1125}$|$0.1772$|$\underline{0.1358}$|$0.1402$|$0.1874$|$0.1923$|$0.1527$|$4$
> > > > > > > $0.25$|$\mathbf{0.1278}$|$0.2112$|$\underline{0.1574}$|$0.1615$|$0.2100$|$0.2071$|$0.1836$|$4$
> > > > > > > $0.3$|$\mathbf{0.1407}$|$0.2169$|$\underline{0.1617}$|$0.1879$|$0.1952$|$0.2285$|$0.1951$|$4$
> > > > > > >
> > > > > > > The result shows that our proposed model is within the middle range of the robustness evaluation on BA-3Motifs graphs with random noise. This can be attributed to two main possible reasons. Firstly, the random noise represents the inherent corruption found in real-world scenarios and is generally not devastating as the adversarial attack. The introduction of random noise results in a lesser degree of irregularity compared to adversarial attack and thus the ability of our method to capture irregularities in corrupted graphs does not bring robustness improvement significantly. In contrast, our method showcases greater robustness than the baseline models when exposed to adversarial attacks on BA-3Motifs graphs. This is due to the fact that such attacks will destroy the influential motifs and consequently amplify the level of irregularity. Secondly, the synthetic BA-3Motifs graph itself adheres to a more regular and format structure than real-world graphs, which adopts a Barabasi-Albert graph as the base and attaches the base with one of three motifs: house, cycle, and grid. Consequently, the potential for our method to achieve robustness improvement is further constrained by the synthetic structural topology.

---

> > > > > > > > ### Comment · Reviewer_Q8Xb · 2023-08-22
> > > > > > > >
> > > > > > > > Thank you for the additional results. I raised my score. I recommend that the authors include the results and discussion on the robustness metric including the ones in the latest reply in the paper.

---

> > > > > > > > > ### Author Response · Authors · 2023-08-22
> > > > > > > > >
> > > > > > > > > Thanks again for your insightful comments and suggestions which are very important to improve our paper.

---

### Author Rebuttal · Authors · 2023-08-10

Thank all reviewers again for your positive opinion of our paper and your valuable comments that have allowed us to improve the manuscript.

We have made point-to-point response to the comments of each reviewer.

---

### Decision · Program_Chairs · 2023-09-21

**Decision:**

Accept (poster)

**Comment:**

The paper presents a novel method, V-InFoR, to explain Graph Neural Networks (GNNs) in the presence of structural corruptions in graphs. The method uses variational inference and an information bottleneck optimization problem to handle both minor and severe graph corruptions. While reviewers generally appreciate the paper's contributions and experimental validations, concerns were raised about the method's necessity and robustness which the authors addressed in their rebuttals. I highly encourage the authors to summarize and report all these new results in the next revision.